# NEURAL FLUID SIMULATION ON GEOMETRIC SURFACES

**Haoxiang Wang**[1] **Tao Yu**[2*] **Hui Qiao**[1*] **Qionghai Dai**[1]
[1]Department of Automation, Tsinghua University, [2]BNRist, Tsinghua University,
`whx22@mails.tsinghua.edu.cn, ytrock@126.com,`
`{qiaohui, daiqionghai}@tsinghua.edu.cn`

## ABSTRACT

Incompressible fluid on the surface is an interesting research area in the fluid simulation, which is the fundamental building block in visual effects, design of liquid crystal films, scientific analyses of atmospheric and oceanic phenomena, etc. The task brings two key challenges: the extension of the physical laws on 3D surfaces and the preservation of the energy and volume. Traditional methods rely on grids or meshes for spatial discretization, which leads to high memory consumption and a lack of robustness and adaptivity for various mesh qualities and representations. Many implicit representations based simulators like INSR are proposed for the storage efficiency and continuity, but they face challenges in the surface simulation and the energy dissipation. We propose a neural physical simulation framework on the surface with the implicit neural representation. Our method constructs a parameterized vector field with the exterior calculus and Closest Point Method on the surfaces, which guarantees the divergence-free property and enables the simulation on different surface representations (e.g. implicit neural represented surfaces). We further adopt a corresponding covariant derivative based advection process for surface flow dynamics and energy preservation. Our method shows higher accuracy, flexibility and memory-efficiency in the simulations of various surfaces with low energy dissipation. Numerical studies also highlight the potential of our framework across different practical applications such as vorticity shape generation and vector field Helmholtz decomposition.

## 1 INTRODUCTION

Fluids are fascinating but complex to simulate, with applications from aerodynamics and hydrodynamics to special effects in computer animation. Flow on the surface is a challenging problem while the practical usages are essential on the visual effects with foam or bubble (Da et al., 2015; Deng et al., 2022), studies of liquid crystal films (Crowdy & Marshall, 2005; Turner et al., 2010), atmosphere/ocean evolution (Miller et al., 1992; Niiler, 2001) and fluid-solid interaction in robotics (Ruan et al., 2021). The incompressible Euler flow model serves as a valuable simplification of real-world fluid dynamics. This model, characterized by a vector field $\boldsymbol{v}(x,t)$ representing velocity, along with pressure $p(x,t)$ and density $\rho_f$, adheres to the following equations:

$$\rho_f(\frac{\partial \boldsymbol{v}}{\partial t} + \boldsymbol{v} \cdot \nabla \boldsymbol{v}) = -\nabla p \tag{1a}$$

$$\nabla \cdot \boldsymbol{v} = 0, \tag{1b}$$

where $x$ in a surface $\mathcal{S}$.

Two main challenges in solving Eqs. 1 puzzle the researchers: one is enforcing the governing equation on the surface and the other is developing efficient approaches for the time integration (advection) and the divergence-free constraint in Eq. 1b, which is critical to ensure the conservation of fluid volume and energy.

Classical approaches often utilize grids or meshes on surfaces and reduce the problem to 2D scenarios. However, they encounter significant challenges in the geometry and the differential operator

---

*Corresponding authors.

computation. Accurate calculation on surfaces relies on high mesh/grid quality, leading to limited robustness and flexibility on different geometry representations. Moreover, the introduction of mesh or grid through spatial discretization also hinders simulation in a continuous spatio-temporal domain owing to the limited memory usage. Finally, the traditional methods need to conduct advection and pressure projection for the divergence-free field, causing the energy dissipation problem. While many alternatives are proposed to solve the problems (Qu et al., 2019; Elcott et al., 2007b; Nabizadeh et al., 2022; Yin et al., 2023), they often come with implementation complexities and lack adaptability to different geometries.

As a promising alternative, simulations based on the neural implicit representations are proposed in recent years (Richter-Powell et al., 2022; Chen et al., 2023). Unlike other data-driven simulation methods (Morimoto et al., 2021; Pfaff et al., 2020) that have limited generalization ability, these methods leverage neural networks to parameterize spatial functions and support the simulation on the continuous domain with limited storage. However, existing methods (Raissi et al., 2019; Chen et al., 2023) can not guarantee the divergence-free property and suffer from the advection error, which leads to the energy dissipation problem. Furthermore, extending these methods to surfaces presents additional challenges for practitioners.

To tackle the challenges, we propose a neural flow on surfaces method based on the neural implicit representation. Neural implicit representation keeps high memory efficiency and supports robust and accurate differential operator computation for the continuous simulation across various geometry representations. Our method leverages a construction on the surface with Closest Point Method (Ruuth & Merriman, 2008) and differential forms, and automatically satisfies the divergence-free constraint, assisting us to enforce the constitutive laws on the surface. To mitigate the challenges of energy dissipation encountered in both classical and advanced methods, we adopt a covariant-derivative based advection to enforce the dynamics of the incompressible fluid. By integrating this process with our divergence-free field construction, our framework eliminates the need for velocity advection and pressure projection, thus minimizing energy dissipation. Furthermore, our framework is versatile and applicable to various tasks such as generation and field decomposition, offering an end-end solution that capitalizes on the advantages of neural representation. In summary, we make the following contributions:

- We present a novel neural physical simulator for surface flow, named NFFS (Neural Functional Flow on Surface), leveraging the Closest Point Method and exterior calculus in the neural implicit representation. Our approach ensures divergence-free properties and adaptability across various geometric surface representations. Notably, it is the first study to present simulation results of incompressible fluid flow on implicitly neural-represented surfaces (Sitzmann et al., 2020) with a guarantee of divergence-free behavior.

- We design a complementary advection process based on the covariant derivatives for fluid dynamics with low energy dissipation.

- We conduct comprehensive numerical studies to verify the correctness, energy preservation, memory efficiency and geometry adaptivity for our proposed framework. Benefitting from the advantages of compact representation, as also highlighted in Chen et al. (2023), our results show that our method achieves approximately **15 times higher accuracy** than other methods with the same storage cost and provides **5 times memory savings** compared to the classic method while accurately describing the phenomenon of fluid dynamics on the surface. Additionally, we demonstrate the conditioning ability of our simulator through an end-to-end generation task and apply it to a real-world velocity field decomposition task.

## 2 RELATED WORKS

**Flow on two-dimensional surface.** The main stream of the fluid simulation consists of the Lagrangian methods like Smoothed Particle Hydrodynamics (SPH) (Gingold & Monaghan, 1977; Monaghan, 1992) and the Euler methods such as stable fluid with Mark-and-Cell grid (Stam, 1999). On surfaces, particle-based methods have been studied extensively in the past decade. The primary focus is the differential operator with SPH-style. Many approximators (Petronetto et al., 2013; Belkin & Niyogi, 2008; Cheung et al., 2015; Nealen, 2004) are proposed and adopted in fluid dynamics (Auer & Westermann, 2013; Leung et al., 2011; Wang et al., 2020; Tao et al., 2022; Suchde, 2021). However, the particle-based methods suffer from high computation cost for the comparable

Figure 1: The paradigm of our paper. Left: the pipeline of our proposed methods. Our method utilizes the implicit neural representation to construct a divergence-free field (showed in Sec. 3.1). We employ exterior calculus and Closest Point Method to construct the surface velocity field $v$ and vorticity field $\omega$ (showed in Sec. 3.2). Subsequently, we adopt the covariant derivatives based advection to calculate the flow in each discretized time iteratively (showed in Sec. 4.1). Right: the potential applications of our proposed method: (a) Our method supports simulation on different surface representation, like analytic surfaces (showed in Sec. 5.1), explicitly represented mesh surface (showed in Sec. 5.2) and implicitly represented surface (showed in Sec. 5.2). (b) Our method also enjoys the ability of conditioning and generation benefitting from our network architecture (showed in Sec 4.2 and 5.3). (c) We demonstrate the effectiveness of our method for the Helmholtz decomposition and analyze the potential usage for scientific research (showed in Sec. 5.4).

accuracy. Dealing with differential forms on surfaces for divergence-free projection also remains challenging, particularly without correspondence between particles.

We next focus our discussions on Euler methods, which can be categorized into velocity-based methods and vorticity-based methods. Velocity-based methods, following Stam (1999), utilize the global or local surface parameterizations (Lui et al., 2005; Hegeman et al., 2009; Hill & Henderson, 2016; Yang et al., 2019), which might introduce undesired distortion. Methods like Stam (2003) restrict the problem to the subdivision surfaces to mitigate this issue. Shi & Yu (2004) tackles the problem by directly simulating on general triangle meshes but entails explicit complex computation of flow lines. More recently, Bhattacharya et al. (2019) extends these to unstructured quadrilateral surface meshes. However, these methods require the advection and divergence-free projection, leading to energy dissipation (Nabizadeh et al., 2022). While there are approaches to address the energy problem such as Mullen et al. (2009); Pavlov et al. (2011) and recent Qu et al. (2019); Deng et al. (2023), they do not primarily focus on the surface dynamics and often entail additional complex computations (such as accurate characteristic mapping (Wiggert & Wylie, 1976)).

For the stream of vorticity-based approaches, methods are grounded in differential forms and exterior calculus on surfaces, circumventing surface parameterization and naturally enforcing the divergence-free property. One of the pioneering methods in this domain was proposed by Elcott et al. (2007b). While their method preserves circulation (vorticity), it suffers from the issues of numerical instability. Another notable vorticity-based approach is presented by De Witt et al. (2012), leveraging eigenfunctions of the Laplacian operator to reduce the computational cost of the Poisson solver when computing velocity from vorticity. This approach has been further extended in works such as Cui et al. (2018; 2021) for spectral-based simulations. However, such methods can be computationally expensive and require numerous eigenvectors for flows with high spatial frequencies. A method closely related to ours is Functional Fluids on Surfaces (Azencot et al., 2014), which employs the Discrete Exterior Calculus on the surface to derive the vorticity and advect with covariant derivatives. This approach achieves convenient computation on surfaces with the energy preservation. Our method builds upon this by employing continuous exterior calculus to generate divergence-free fields and vorticity functions on surfaces. The theoretical accuracy of our method is higher with a sufficient number of surface samples. Moreover, we incorporate the neural implicit representation to alleviate the high memory burden associated with high smoothness and accuracy requirements.

**Physical Simulation based on Neural Network.** Neural physical simulation can be divided into two main streams. The first one is the data-driven simulation. This type of methods often aims at solving simulation problems based more on data but less on the governing equation. They often adopt the training data from the classical solvers or the real world observation to make the neural network learn the physical rules and generalize it to other scenarios. Some convolution network approaches (Morimoto et al., 2021) and designed U-Net approaches (Lu et al., 2019) show higher efficiency than the classical solvers. Neural operator approach (Li et al., 2020) makes full use of

the Fourier layer and becomes an important milestone for the type of methods. Recently, a newly proposed Lagrangian Flow Networks (Torres et al., 2023) embeds the idea of the characteristic mapping and design a data-driven PDE solver. However, these methods may struggle to generalize well to different initial/boundary conditions, material parameters, or geometries. The training data acquisition and time-consuming training processes also block their wide applications. Another stream of the line is to embed the governing equations into the network. For this type, one representative direction we have to mention is Physical-Inform Neural Network (PINN) (Raissi et al., 2019). The method designs the physical loss term according to the governing equation and the neural network is trained to extract the features of the spatiotemporal correlation and fit the target field via the physical loss. However, the ways to directly force the neural network to fit all the physical rules make the training process difficult. The training will cost a very long time and often can not achieve the required accuracy. To fulfill this issue, the implicit neural representation Chen et al. (2023) is also introduced to better describe the spatiotemporal dependencies and reduce the burden of network training. Nevertheless, this method enforces physical laws via an operator splitting manner, leading to the energy dissipation problem. Kim et al. (2019); Rao et al. (2020); Richter-Powell et al. (2022) propose divergence-free neural field construction approaches and optimize the advection process. These methods, while effective in guaranteeing the divergence-free property, face challenges in direct application to surface flows and encounter difficulties in optimization processes.

Hence, in this work, we propose a novel framework that constructs a neural-represented divergence-free vector field on surfaces to embed the efficient spatial representation with the physical prior. We also design the corresponding covariant derivative based advection process for the fluid dynamics computation. Our method can achieve a memory-efficiency, accurate and energy-preserving simulation on different surfaces representation, even for the implicit neural represented surfaces. We show the paradigm of our paper in Fig. 1.

## 3 NEURAL FLOW ON SURFACES

In this section, we present a framework designed to enforce the divergence-free characteristics for vector fields on surfaces. We first present the general philosophy to construct the divergence-free field then we explain how we apply it to the different surface representations, especially in implicit neural represented surfaces.

### 3.1 CONSTRUCTION OF THE DIVERGENCE-FREE VECTOR FIELD

We adopt the terms of differential forms to derive the divergence-free vector field. More preliminaries are shown in Do Carmo (1998) and Appendix A. in Richter-Powell et al. (2022). We first discuss the divergence-free vector field on a Riemannian manifold $\mathcal{M}$, (e.g. $\mathcal{R}^3$).

Let $\mathcal{A}^k(\mathcal{M})$ as the space of $k$-forms. The operator $d$: $\mathcal{A}^k(\mathcal{M}) \to \mathcal{A}^{k+1}(\mathcal{M})$ is the exterior derivative while $*$: $\mathcal{A}^k(\mathcal{M}) \to \mathcal{A}^{n-k}(\mathcal{M})$, where $n$ is the dimension of $\mathcal{M}$, denotes the Hodge star operator mapping each $k$-form to an $(n-k)$-form. Our vector field can be represented as 1-form, $\boldsymbol{v} = \sum_{i=1}^{n} v_i dx_i$.

With the notation of the differential forms and the computation tools of the differential manifold, we can define the divergence $\text{div}(v) = *d*\boldsymbol{v}$. We can observe that $\text{div}(v)$ can be expressed with differential k-forms, but after computation it reduce to a 0-form, resulting in a scalar function.

A fundamental property of the exterior calculus need to be proposed that for an arbitrary $(n-2)$-form $\mu \in \mathcal{A}^{n-2}(\mathcal{M})$, we have

$$d^2\mu = d(d\mu) = 0, \tag{2}$$

then it follows that

$$\boldsymbol{v} = *d\mu \tag{3}$$

is divergence-free since $**$ yields a sign function. Consequently, our objective is to construct a parametric $(n-2)$-form $\mu$ that enforces a divergence-free $\boldsymbol{v}$. We can construct a network parameterization to construct $\mu$ and derive the required velocity field $\boldsymbol{v}$ with the divergence-free property for the incompressibility via Eq. 3. Therefore, the main issue comes to the computation of the operator $*$ and $d$ on our provided surface $\mathcal{S}$. We calculate it by Closest Point Method in the following subsection.

### 3.2 CONSTRUCTION OF THE NEURAL FLOW ON SURFACES

On the surfaces $\mathcal{S}$ embedded in $\mathcal{R}^3$, the analysis differs. To facilitate a more convenient analysis of differential forms on the surface without specific surface parameterization, we consider studying $\mathcal{S}$ in $\mathcal{R}^3$ instead of $\mathcal{R}^2$. The Closest Point Method (CPM) (Ruuth & Merriman, 2008; Li et al., 2023a) serves as a tool to transform differential forms on surfaces into ones defined in $\mathcal{R}^3$, using the closest point on the surface. We define the inclusion map $j : \mathcal{S} \to \mathcal{N} \subset \mathcal{R}^3$, where $\mathcal{N} \subset \mathcal{R}^3$ is a neighborhood of $j(\mathcal{S})$. The closest point function $cp : \mathcal{N} \to \mathcal{S}$ takes a point in the neighborhood and returns the closest point on the surface. Then we can define the pullback operator $j^* : \mathcal{A}^k(\mathcal{N}) \to \mathcal{A}^k(\mathcal{S})$ and $cp^* : \mathcal{A}^k(\mathcal{S}) \to \mathcal{A}^k(\mathcal{N})$ to map the tangential vector in the neighborhood and on the surfaces. The endomorphism $(j \circ cp)^* = cp^* j^* : \mathcal{A}^k(\mathcal{N}) \to \mathcal{A}^k(\mathcal{N})$ replaces a neighbor $k$-form by its extension from its value at the surface $j(\mathcal{S})$.

With the Closest Point Method, we can construct the divergence-free field on the surfaces $\mathcal{S}$ in the following theorem. [1]

**Theorem 3.1.** *Given a parameterized scalar function (stream function) $\sigma : \mathcal{S} \to \mathcal{R}$, one can construct divergence-free $\boldsymbol{v} : \mathcal{S} \to \mathcal{R}^3$, for $x \in \mathcal{S}$:*

$$\boldsymbol{v}(x) = j^*((\nabla(cp^*\sigma) \circ j(x)) \times \boldsymbol{n}(x)), \tag{4}$$

*where $\boldsymbol{n} : \mathcal{S} \to \mathcal{R}^3$ represents the normal of the surfaces, and the corresponding vorticity function $\omega : \mathcal{S} \to \mathcal{R}$ can be defined as*

$$\omega = (\nabla \times \boldsymbol{v}(x)) \cdot \boldsymbol{n}(x). \tag{5}$$

Equation 4 defines a stream function and employs the gradient operator to transform into a vector field. The operator $cp^*$ pulls the vector field back into the ambient space, while $\boldsymbol{n}$ and $j^*$ ensure that the field lies on the tangent plane and is restricted to the surface. The vorticity can be interpreted as having a rotation axis aligned with the surface normal, as it is evaluated directly on the surface, making it a scalar field. To provide a clearer understanding of the computation described in Theorem 3.1, we include an illustration in Fig. 2.

**Proof sketch:** Our goal is to construct a divergence-free field. We adopt the form in Sec. 3.1 $*d\mu$ as the basis. Then we calculate the value of this differential form for a surface function. We utilize the Closest Point Method and pullback the surface field $\boldsymbol{v}$ into the ambient space. We extend the differential form with surfaces normals to $\mathcal{R}^3$ by the pullback and derive the parameterization on $\mathcal{R}^3$ that preserves the divergence-free. Moreover, the pullback utilizes the closest point and shares the property that the closest point of the surface point is the point itself. Therefore, we can simply constrain this parameterization for $\boldsymbol{v}$ on the surface and derive our required surface field that satisfies the divergence-free property on the surface. We can also derive vorticity function as $*d\boldsymbol{v}$ through the similar process.

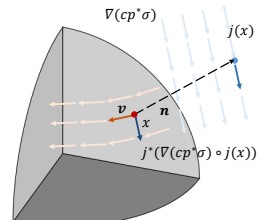

Figure 2: Illustration for divergence-free field.

**Remark:** The construction of the divergence-free field in Eq. 4 is related to the term "surface curl" in the electromagnetic's Helmholtz decomposition (Scharstein, 1991). However, our theorem provides a more formal formulation indicating why it is divergence-free from the perspective of the Closest Point Method and adopts it as a neural parametric function for surface vector field dynamics. Building on our analysis, we can further derive a scalar vorticity function on the surface to support the fluid dynamics simulation.

Following the conclusion of Theorem 3.1, we can construct our neural flow (a parametric vector field based on $\sigma$ and $\boldsymbol{n}$) for different representations of the surfaces. For the explicit representations like sphere, plane (analytical), and mesh (discretized), our neural flow can be constructed with sampling on the surfaces and computing/querying the normal. Moreover, the most significant aspect of our construction lies in its applicability to implicit representations, particularly the implicit neural representation (INR), such as DeepSDF (Park et al., 2019) and siren (Sitzmann et al., 2020). These methods take $x \in \mathcal{R}^3$ as input and return the sign distance function $s(x)$. In the application on this

---

[1]Here we follow the similar discussions as Richter-Powell et al. (2022) and omit the case for non-zero homology for clear and concise in theoretical analysis. We actually can address the issue as we state in the following sections.

type of surface, points $x \in \mathcal{S}$ can be simply sampled by $x - s(x)\frac{\nabla s(x)}{\|\nabla s(x)\|_2}$ leveraging the characteristics of the sign distance function, while the normal on the surface can be computed using $\nabla s(x)$. Then, for the sampled points on the surface, we can create a parametric form $\sigma(\theta, x)$ and finalize the divergence-free construction with the normals following Eq. 4. The function can be also derived via Eq. 5.

Our construction of $v$ directly enables tasks like Hodge-Helmholtz decomposition (extracting the divergence-free component given a vector field). It can be simply achieved with our construction with the mean square error loss for $v$. In addition, the construction of $\omega$ can be utilized to conduct advection serving for the vortices dynamics in fluid simulation, as we will introduce subsequently.

## 4 APPLICATIONS WITH NEURAL FLOW ON SURFACES

In this section, we put our construction of neural flow on surfaces into practice. We first discuss the advection for the simulation of the real fluid dynamics. Then we delve into applications involving the conditioning.

### 4.1 ADVECTION OF NEURAL FLOW ON SURFACES

Our vector field inherently satisfies the incompressibility condition with the divergence-free construction, eliminating the need for pressure projection to enforce the constraint and associated errors. However, our proposed field, while possessing this advantageous property, introduces a new challenge: how can we advect this neural field over time to adhere to physical laws? The incompressible Euler equation can be written with the vortex form (Davidson, 2015):

$$\frac{\partial \omega_t}{\partial t} = -(\boldsymbol{v}_t \cdot \nabla)\omega_t, \qquad (6)$$

where $\boldsymbol{v}_t$ is the divergence-free velocity field and $\omega$ is the vorticity field.

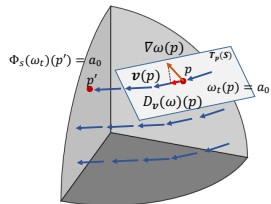

**Remark:** A direct sequence of Eq. 6 is that $\omega_t$ is transported in the same manner as fluid particles. This implies that we can advect/diffuse $\omega_t$ similarly to a scalar property carried by the particles, such as temperature.

Figure 3: Illustration for covariant derivative.

For the advection process, instead of the semi-Lagrangian scheme (Staniforth & Côté, 1991) that requires the discretization, we opt for a functional way to fully enjoy the continuity of our neural field construction (Azencot et al., 2014). As indicated in the remark above, we define the flow $\phi_t(p)$ that denotes the particle position after time $t$ for the particle that starts at time 0. That is,

$$\frac{\partial \phi_t(p)}{\partial t} = \boldsymbol{v}_t(\phi_t(p)), \quad \phi_0(p) = p, \qquad (7)$$

for $p \in \mathcal{S}$. Then we can observe $\phi_t$ is an invertible self-map on $\mathcal{S}$ and can be adopted to transport $\omega_t$. Define $\phi_t$ acts on the smooth function $f : \mathcal{S} \to \mathcal{R}$ through the push forward:

$$\Phi_t(f) = f \circ \phi_t^{-1}. \qquad (8)$$

We can find out our vorticity mapped by $\Phi$ satisfies:

$$\omega_t = \Phi_t(\omega_0). \qquad (9)$$

Next we implement the advection of $\omega$. We discretize time and consider time $i$, $\omega_i$ with the time step $h$. Then we can impose the following requirement, by assuming $\boldsymbol{v}_i$ advecting for time $h - t$ and $\boldsymbol{v}_{i+1}$ advecting for time $t$:

$$\omega_{i+1} = \Phi_{i \to i+1}(\omega_i) = \Phi_t^{\boldsymbol{v}_{i+1}} \circ \Phi_{h-t}^{\boldsymbol{v}_i}, \qquad (10)$$

as $t \in [0, h]$, which is similar with the implicit Euler scheme (Butcher, 2016). This enables us to derive the equivalence for the forward advected $\omega_i$ (along $\boldsymbol{v}_i$) and backward advected $\omega_{i+1}$ (along $\boldsymbol{v}_{i+1}$) by taking $t = h/2$ as follows:

$$\Phi_{-h/2}^{\boldsymbol{v}_{i+1}}(\omega_{i+1}) = \Phi_{h/2}^{\boldsymbol{v}_i}(\omega_i). \qquad (11)$$

Then our goal turns to compute $\Phi_t^{\boldsymbol{v}}$. We first define the covariant derivative $D_{\boldsymbol{v}}(f)$ as a function $g$, which measures the change in $f$ w.r.t. the flow under $\boldsymbol{v}$:

$$g(p) = D_{\boldsymbol{v}}(f)(p) = \lim_{t \to 0} \frac{f(\phi_t(p)) - f(p)}{t}. \tag{12}$$

A classic result in Riemannian geometry is that the covariant derivative can be computed as Morita (2001) (as shown in Fig. 3):

$$D_{\boldsymbol{v}}(f)(p) = g(p) = \langle (\nabla f)(p), \boldsymbol{v}(p) \rangle_p. \tag{13}$$

With the conclusion in Azencot et al. (2013) (Lemma 2.5), we can derive that

$$\Phi_t^{\boldsymbol{v}}(\omega) = \exp(tD_{\boldsymbol{v}})\omega = \sum_{k=0}^{\infty} \frac{(tD_{\boldsymbol{v}})^k \omega}{k!}, \tag{14}$$

with respect to $\boldsymbol{v}$. Using Eq. 13 and Eq. 14 above, along with the first-order approximation estimated by inner product, we can derive the following expression for each time step:

$$\mathcal{L}_i = \omega_i - \omega_{i+1} + \frac{h}{2} \langle \nabla \omega_i, \boldsymbol{v}_i \rangle + \frac{h}{2} \langle \nabla \omega_{i+1}, \boldsymbol{v}_{i+1} \rangle = 0. \tag{15}$$

Then our advection schemes involve iteratively minimizing the loss function $\mathcal{L}_i$ with respect to parametric $\omega_{i+1}$ and $\boldsymbol{v}_{i+1}$ through Theorem 3.1 and preparing them for the advection for next time $i + 2$. More specifically, for the parameters $\theta_i$ of the parametric function $\omega_i$ and $\boldsymbol{v}_i$, we seek to optimize

$$\theta_{i+1} = \operatorname{argmin}_{\theta_{i+1}} \sum_{x \in \mathcal{M} \subset \mathcal{S}} \mathcal{L}_i(h, \omega(\theta_i), \boldsymbol{v}(\theta_i), \omega(\theta_{i+1}), \boldsymbol{v}(\theta_{i+1})), \tag{16}$$

where $\mathcal{M}$ is the sample set from the surface and $\{\theta_i\}_{i=0}^T$ represents the vector field for $T$ time steps.

For the initial time $\theta_0$, we utilize the given velocity field $\boldsymbol{v}_0$ or vorticity field $\omega_0$ and conduct fitting for initialization. For the case that $\boldsymbol{v}_0$ and $\omega_0$ are all provided, these exists the harmonic component that does not contribute to the vorticity but influences the velocity field. The term is often associated with the topological structure and modeled as time-invariant (Azencot et al., 2014). Therefore, we need to construct another time-invariant field parameterization (MLP) to fit the harmonic term for the non-zero homology. We can follow Richter-Powell et al. (2022); Azencot et al. (2014) and simply add another parameterized vector field $\eta$ to Eq. 4. Specifically, at the initial time, we employ another MLP $\eta$ to learn the residual in the initial velocity after fitting the vorticity, treating this residual as the harmonic components. Similar with Azencot et al. (2014), $\eta$ remains time-invariant in subsequent velocity computations and not trained in the following iterative computation. More discussions about the topology are included in Appendix F.1.

For all the examples presented in this work, we solve this time-integration optimization problem via Adam (Kingma & Ba, 2014), a first-order stochastic gradient descent method. The computational process is showed in the pseudocode Algorithm 1 in Appendix D.

## 4.2 CONDITIONING PROPERTY IN NEURAL FLOW ON SURFACES

As mentioned in Theorem 3.1, we need to construct parametric $\sigma(\theta, x)$ for $x \in \mathcal{S}$ and parameters $\theta$. The formulation also allows us to conduct conditioning, i.e. function $\sigma(\theta, x, z)$, where $z$ is the features of the conditioning for the divergence-free field. This property can be leveraged in different tasks. For instance, in Sec. 4.1, the feature $z$ can represent time $t$. Moreover, for the tasks involving the vector field generation, encode $z$ can represent semantic features extracted from natural scenes or images, thereby enabling control over the shape, scale or other more semantic information of the field. We further present an example utilizing the variation auto-encoder for the vorticity field generation (settings in Appendix C and results in Sec. 5.3), showing the capability of our framework to involve the semantic prior.

## 5 NUMERICAL STUDIES

In this section, we conduct numerical studies for our proposed framework. Our primary emphasis lies in verifying the efficacy of our framework in fluid dynamics across various surface representations, exploring conditioning characteristics, and demonstrating practical applications such as the

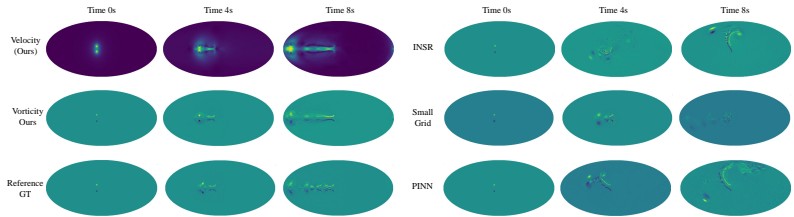

Figure 4: Comparison results for sphere jet flow.

Helmholtz decomposition using real-world data. It is worth noting that our method does not rely on any training data, distinguishing it from certain neural network-based simulation methods (Morimoto et al., 2021; Lu et al., 2019; Pfaff et al., 2020; Torres et al., 2023).

| Methods | Error | Time | Storage |
|---------|-------|------|---------|
| PINN | 1.73e5 | 12.1 h | 568.1KB |
| INSR | 8.63e4 | 20.2 h | **516.3KB** |
| Small-F.S. | 5.34e3 | **0.8h** | 583.8KB |
| Ours | **2.89e2** | 16.5 h | 532.8KB |
| GT | N/A | 8.3 h | 2643.0 KB |

Table 1: Quantitative results for the sphere jet flow. Error: mean square error (MSE) averaged by 100 time steps on 81924 mesh vertices.

### 5.1 Flow on the analytical surfaces

In this subsection, we present the benchmark studies for our proposed framework on the analytical sphere and inclined plane, which allows us to easily derive samples and normal. We design the sphere jet flow and inclined plane to assess the correctness of our methods, compared against functional fluid dynamics and other baseline methods. Additional, we also utilize a sphere rot case to further validate the energy-preservation characteristics of our method in Appendix E.1.

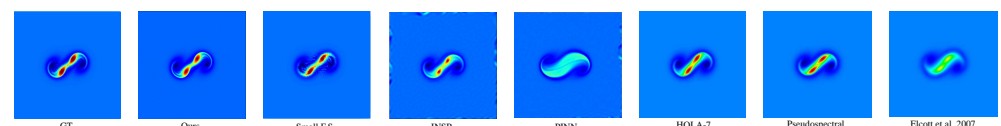

Figure 5: Vorticity comparison for inclined planar Taylor vortices on the 40th time step. HOLA; Pseudospectral; Elcott et al 2007 results are quoted from McKenzie (2007).

**Sphere Jet.** On the sphere, we also simulate a jet flow as illustrated in Fig. 4. We initialize two opposite vortices on the sphere to generate the jet. Our vorticity results are compared with those from PINN (Raissi et al., 2019), Implicit Neural Spatial Representations (INSR) (Chen et al., 2023) and Functional Fluid on Surfaces with the same storage cost. We use higher-resolution Functional Fluid on Surfaces as the reference ground truth, whose vector field storage is 5 times than ours. The results depicted in Fig. 4 indicate our method better characterizes the jet phenomenon of the flow vortices compared with other methods. Our method allows optimization on the subspace of divergence-free functionals via the physical constraint in network design while others need projection to the divergence-free functional subspace via an extra-designed loss function, which brings the error in each discretized time step, leading to the significant inaccuracy with the cascading effects in Fig. 4. Additionally, our method produces smoother results since we sample across the entire sphere rather than only solving for the mesh vertices as in high/low-resolution Functional Fluid on Surfaces. In other words, for higher resolution and smoother results, the Functional Fluid on Surfaces method requires more storage, whereas our memory cost remains constant. We provide the quantitative comparison for the vorticity values on the reference ground truth method's mesh vertices. The mean square error in 100 time steps is shown in Tab. 1. The results demonstrate that at the same memory cost, ours can achieve the highest accuracy compared to other methods. More empirical results and comparison with both classical and advanced methods are included in Appendix E.3 to further verify the effectiveness of the proposed framework.

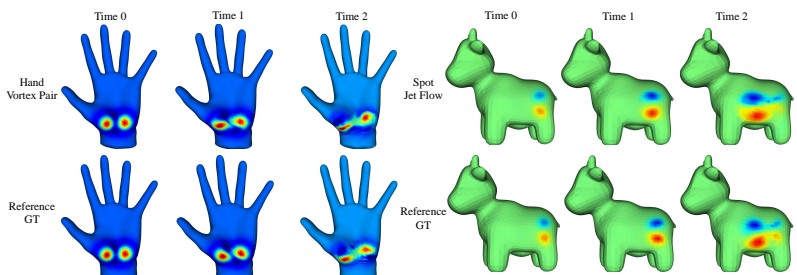

Figure 6: Results for flow on explicit meshes.

**Taylor vortices on inclined plane.** We also simulate Taylor vortices on an inclined plane with a known normal. The phenomena observed should be similar to those on a 2D plane. The initialization of the vortices is referred to McKenzie (2007). The quantitative results are displayed in Fig. 5. Ours capture the details of vortices phenomenon with high smoothness and low alias. The classical methods with the close memory cost lost some details or leaves aliasing artifacts. The advanced method INSR shows a similar result but still fails in details and vortices energy preservation due to the energy dissipation. PINN suffers from the largest energy dissipation and the result is the least accurate due to the enforcement of the divergence-free constraint. We also include the quantitative results and more comparison with classic solvers in Appendix E.2 to further show the memory efficiency and accuracy of our proposed method.

## 5.2 FLOW ON THE EXPLICIT AND IMPLICIT MESHES

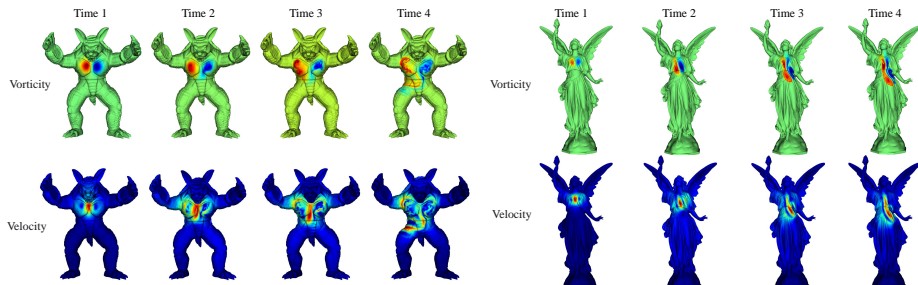

Figure 7: Results for flow on implicit neural representation.

In this subsection, to further demonstrate the generality and robustness of our methods, we present the flow results for both explicit and implicit geometry. For these geometries, the samples and normal can not be analytically derived, resulting in inherent errors. Nevertheless, we demonstrate that our method still performs effectively and captures the flow phenomenon accurately[2].

**Flow on the explicit meshes.** We initialize the Taylor vortices pair and jet flow on the explicit hand (Jacobson et al., 2018) and spot model (Crane et al., 2013a). The results are depicted in Fig. 6. They show that albeit less smooth results, due to the relatively low mesh resolution, our flows exhibit the similar behavior as the reference GT and effectively capture the vortices and jet phenomenon.

**Flow on the implicit neural representation.** We also simulate a jet flow on the implicit neural represented Armadillo (Krishnamurthy & Levoy, 1996) and Lucy (Turk & Levoy, 1994). The results in Fig. 7 accurately capture the smooth jet flow phenomenon on different implicit surfaces. Surprisingly, the classic functional flow on surface method fails to converge for both the mesh obtained from marching cubes on our implicit neural representation and even the original mesh (using a Newton solver). To substantiate our claim regarding non-convergence, we include more studies in the supplementary (Appendix E.4) that lists the simulation crash time steps across different marching cube resolutions for the traditional methods. The outcome indicates that our method keeps higher robustness than the traditional method which depends on mesh quality and exhibits instability but consumes high memory when applied to the implicit neural representations. Instead, our method

---

[2]Note that we do not compared with INSR and PINN in these cases, since the advection and projection in the two methods can not be simply adopted to the flow on the various surfaces without the surface parameterization to $\mathcal{R}^2$ and the continuous pullback to $\mathcal{R}^3$, which are beyond the scope of the work.

only needs the samples in 3D rather than meshes, thereby avoiding the subtle in the geometry and topology inherent in the complex shape, and therefore derive the smooth jet flow results. Moreover, the learnable neural divergence-free field is capable of tolerating errors and supports more robust simulation in the long term. Generally speaking, our method supports wider applications on different geometry representations with high memory efficiency in practice.

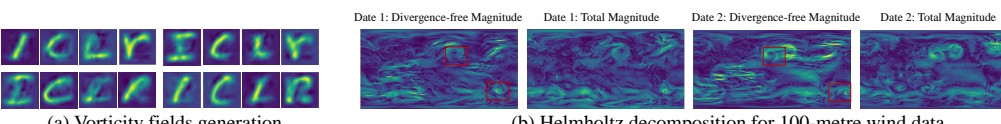

(a) Vorticity fields generation.         (b) Helmholtz decomposition for 100-metre wind data.

Figure 8: Vorticity fields generation and decomposition tasks.

### 5.3 FLOW WITH CONDITIONING

To verify the conditioning property for our proposed neural-based framework, we design the generation task as stated in Sec. 4.2 and Appendix C. We take EMNIST dataset (Cohen et al., 2017) as the image input and generation the divergence-free velocity fields with the vorticity imitating the silhouettes of alphabets.

We construct a variational auto-encoder for the vorticity fields representing different alphabets in Fig. 8 (a). The numerical studies demonstrate the feasibility of our proposed framework, benefitting from the neural network, to effectively utilize conditioning encodes from other modalities. Ours operates in a more end-to-end manner compared to approaches that first generate and then fit with classic simulators, thereby avoiding the repeated fitting process for each generation.

### 5.4 FLOW FOR HELMHOLTZ DECOMPOSITION

Finally, we apply our method to the real-world atmosphere dataset (Raoult et al., 2017). The Helmholtz decomposition is performed on the 100-metre wind velocity data with our proposed framework. In this study, we make the sphere assumption for the latitude and longitude coordinates in the dataset and derive the normal analytically. The results, shown in Fig. 8 (b), reveal identifiable vortices after the decomposition. Further inference can be made regarding atmosphere information from the results (Cao et al., 2014; Hammond & Lewis, 2021).

## 6 DISCUSSION AND CONCLUSION

This work adopts the neural representation to construct the parametric divergence-free vector field based on the Closest Point Method that supports exterior calculus on the surfaces. Our framework facilitates field construction along with covariant-derivative-based advection directly on various surface representations, especially on the implicit neural representation, bypassing the need for marching cubes or meshing functions. It is not well-supported by classic simulators. Our framework aligns well with current trends in neural implicit representation methods like DeepSDF (Park et al., 2019) and NeuS (Wang et al., 2021) from this point. The experiment results validate the correctness, energy-preserving and memory-efficiency of our method. Furthermore, our framework shows high robustness and flexibility and also supports various conditioning tasks for further applications.

While offering important benefits, our method also suffers from limitations. Our main limitations stem from topology, geometry and the time efficiency, and we have discussed the details in Appendix F. For more future extensions, a theoretical analysis of convergence and stability of our method would be valuable. Also, more efforts in advection can be made, including high order approximation (Suzuki, 1985), combination with other advanced methods like reference mapping (Li et al., 2023c) and data-driven network (Liu et al., 2021). Expanding support for boundary conditions and flow viscosity [3] on the surfaces are also vital, for more realistic and practical applications. It is also a promising direction to integrate our proposed framework into the existing neural reconstruction pipeline like NeRF (Mildenhall et al., 2021; Wang et al., 2021), enabling more improved performances in dynamic reconstruction and inverse physics with the vision input.

---

[3]The viscosity term can be also supported by applying the Laplacian $\Delta$ to the vorticity function $\omega$ and $\langle(\nabla\omega), \boldsymbol{v}\rangle$. We can derive the calculation on the surface using Theorem 3.1 by substituting the stream function with $\omega$ and $\langle(\nabla\omega), \boldsymbol{v}\rangle$.

ACKNOWLEDGEMENTS

This work was supported by National Key R&D Program of China (No.2023YFB3209700 and No.2024YFB2809101), National Natural Science Foundation of China (No.62322110 and No.62171255).

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

## A    PRELIMINARIES FOR THE MATHEMATICAL TOOLS

### A.1    THE DIFFERENTIAL GEOMETRY IN $\mathcal{R}^n$

We provide an in-depth discussion of our field construction which follows the introduction in Appendix A of Richter-Powell et al. (2022). Please refer to the work for more details. Readers with a background in differential geometry can skip this section. We first discussion the basic concept in the differential geometry for the introduction of differential form that supports us to derive and prove our theorem in the paper.

We take a local coordinate chart for $x \in \mathcal{R}^n$ as $x = (x_1, ..., x_n)$ and $dx_1, .., dx_n$ denotes the coordinate differentials, i.e. $dx_i(x) = x_i$, $i \in [n] = \{1, ..., n\}$, which is also the co-vector field of the local coordinates. Note that we discuss $\mathcal{R}^n$ as an example and all the definition can be extended for smooth manifold and well-defined but needs the construction of the local chart or other mathematical manipulations. For more extensive introduction see Do Carmo (1998); Morita (2001).

Define the linear vector space $*^k(\mathcal{R}^n)$ for $\mathcal{R}^n$ as the space of the $k$-linear alternating map:

$$\phi : \overbrace{\mathcal{R}^n \times \ldots \mathcal{R}^n}^{k \ times} \to \mathcal{R}. \tag{17}$$

A $k$-linear alternating map $\phi$ is linear in each coordinate and satisfies the alternating property:

$$\phi(v_1, \ldots, v_i, \ldots, v_j, \ldots, v_n) = -\phi(v_1, \ldots, v_j, \ldots, v_i, \ldots, v_n). \tag{18}$$

The basis of the space $\Lambda^k(\mathcal{R}^n)$ can be denoted with the differentials by $dx_{i_1} \wedge \cdots \wedge dx_{i_k}$. The way of the basis act on $k$-vectors, $v_1, ..., v_k \in \mathcal{R}^n$ as:

$$dx_{i_1} \wedge \cdots \wedge dx_{i_k}(v_1, ..., v_k) = \frac{1}{k!} \sum_{\Upsilon \in S_k} sgn(\Upsilon) dx_{i_1} \wedge \cdots \wedge dx_{i_k}(v_{\Upsilon(1)}, ..., v_{\Upsilon(k)}) = det[dx_{i_r}(v_s)]_{r,s \in [k]}, \tag{19}$$

where $S_k$ is a permutation for $i_1, ..., i_k$.

More specifically, for $\chi \in \Lambda^k(\mathcal{R}^n)$ can be represented by the basis as:

$$\chi = \sum_{i_1 < i_2 \cdots < i_k} a_{i_1, \ldots, i_k} dx_{i_1} \wedge \cdots \wedge dx_{i_k} = \sum_I a_I dx^I, \tag{20}$$

where $i_1, \ldots, i_k \in [n]$ and $I = (i_1, \ldots, i_k)$ determining scalars $a_I$ and $dx^I = dx_{i_1} \wedge \cdots \wedge dx_{i_k}$.

The space of differential $k$-form $\mathcal{A}^k(\mathcal{R}^n)$ ($k$-forms in the main content) is defined by the smooth function $w_I : \mathcal{R}^n \to R$ as $w \in \mathcal{A}^k(\mathcal{R}^n) : \mathcal{R}^n \to \Lambda^k(\mathcal{R}^n)$:

$$w = \sum_I w_I dx^I. \tag{21}$$

Then the differential operator can be viewed as $d : \mathcal{A}^0(\mathcal{R}^n) \to \mathcal{A}^1(\mathcal{R}^n)$ as:

$$df(x) = \sum_{i=1}^n \frac{\partial f}{\partial x_i}(x) dx_i. \tag{22}$$

The exterior derivative $d : \mathcal{A}^k(\mathcal{R}^n) \to \mathcal{A}^{k+1}(\mathcal{R}^n)$ is defined as a linear operator and can be calculated by:

$$dw(x) = \sum_I dw_I \wedge dx^I, \tag{23}$$

where the calculating rules for the exterior product $\wedge$ for two forms $w = \sum_I w_I dx^I$ and $\iota = \sum_J \iota_J dx^J$ can be derived by (with Eq. 19):

$$w \wedge \iota = \sum_{I,J} w_I \iota_J dx^I \wedge dx^J. \tag{24}$$

For the exterior derivative, an important property is $dd = 0$. It can be checked with the definition Eq. 23 and $\frac{\partial f^2}{\partial x_i \partial x_j} = \frac{\partial f^2}{\partial x_j \partial x_i}$. It can be also extended to the general manifold $\mathcal{M}$ with the local chart and the similar proof.

The hodge operator $* : \mathcal{A}^k(\mathcal{R}^n) \to \mathcal{A}^{n-k}(\mathcal{R}^n)$ maps each $k$-form to $(n-k)$-form by functioning as:

$$*w_I(dx^I) = (-1)^{sgn(\Upsilon)}w_I(dx^J), \tag{25}$$

where $\Upsilon$ denotes the permutation $(I, J) = (i_1, ..., i_k, j_1, ..., j_{n-k})$ on $[n]$. We can also derive the

$$** = (-1)^{k(n-k)} \tag{26}$$

in $\mathcal{R}^n$ (the similar sign function result also holds in the general $\mathcal{M}$). We can also define the codifferential operator $\delta : \mathcal{A}^k(\mathcal{R}^n) \to \mathcal{A}^{n-k}(\mathcal{R}^n)$ as:

$$\delta = (-1)^{n(k+1)+1} * d * . \tag{27}$$

Then for a vector field $v = (v_1, ..., v_n)$, we can represent it as 1-form $\boldsymbol{v} = \sum_{i=1}^n v_i dx_i$. Then the divergence can be calculated as follows:

$$*d * \boldsymbol{v} = *d \sum_{i=1}^n v_i * dx_i = * \sum_{i=1}^n \frac{\partial v_i}{\partial x_i} dx_1 \wedge \cdots \wedge dx_n = \sum_{i=1}^n \frac{\partial v_i}{\partial x_i}, \tag{28}$$

where the equation holds with the fact $dx_i \wedge dx_i = 0$.

Following the discussion above, we can also simply prove that our Eq. 3 is divergence free:

$$*d * \boldsymbol{v} = *d * *d\mu = (-1)^r * dd\mu = 0, \tag{29}$$

where $r = (n-1)$ for $(n-1)$-form $d\mu$ in $\mathcal{R}^n$.

There are various approaches to parameterize $\mu$. One alternative involves directly parameterizing $\mu$ using an antisymmetric matrix-valued function (Richter-Powell et al., 2022) to characterize the antisymmetric property in hodge star and wedge product computation. Also, the codifferential operator $\mu = \delta\nu$ can also be employed to keep the properties required for $\nu \in \mathcal{A}^{n-1}(\mathcal{R}^n)$, and

$$\boldsymbol{v} = *d\delta\nu. \tag{30}$$

This form can be written as a vector representation with $\nu$ and simplify the parameterization.

We also propose a specific example to explore the relationship between the divergence-free field construction through the differential operator and the classic approach involving the curl operator (vector potential in fluid simulation) to enforce divergence-free.

We take $\mathcal{R}^3$, $\mu = \nu_x dx + \nu_y dy + \nu_z dz$ for $\nu = (\nu_x, \nu_y, \nu_z)$ and the differentials $dX = (dx, dy, dz)$. Then we can derive

$$\boldsymbol{v} = \left(\frac{\partial \nu_x}{\partial y} - \frac{\partial \nu_y}{\partial x}\right) dz + \left(\frac{\partial \nu_y}{\partial z} - \frac{\partial \nu_z}{\partial y}\right) dx + \left(\frac{\partial \nu_x}{\partial z} - \frac{\partial \nu_z}{\partial x}\right) dy = *d\mu = (\nabla \times \nu) \cdot dX$$
$$= (\text{curl } \nu) \cdot dX. \tag{31}$$

With Eq. 3, we can deduce that $\boldsymbol{v} = *d\mu$ is divergence-free. Additionally, it coincides with the curl operator of the vector $\nu$, which happens to be a vector field in $\mathcal{R}^3$. Consequently, the curl of a vector field possesses zero divergence (div $\circ$ curl) in $\mathcal{R}^3$. This conclusion is frequently utilized in simulations, known as the vector potential design (Elcott et al., 2007a; Chang et al., 2022), where the divergence-free field construction leverages curl operator. However, for the general manifold, it might be tricky to define each "curl" operator and the classical vector potential can not even be simply designed as a vector, while the differential operator computation for the divergence-free field construction can still work. We will next focus on the calculation of $*d$ operator for the divergence-free parameterization.

## A.2 Introduction to the Closest Point method

We also discuss some preliminaries for the Closest Point Method. We follow the introduction in King et al. (2023). For more details, please refer to King et al. (2023). Consider $\mathcal{S}$ embedded in $\mathcal{R}^3$. The Closest Point Method utilizes a closest point surface representation, which is a mapping from $x \in \mathcal{R}^3$ to the point $cp(x) \in \mathcal{S}$. The point $cp$ is defined as the closest point on $\mathcal{S}$ to $x$ in Euclidean distance, i.e.

$$cp(x) = \text{argmin}_{y \in \mathcal{S}} \|x - y\|. \tag{32}$$

For the smooth surfaces, $cp$ is unique and well-defined in a sufficiently narrow tubular neighborhood $\mathcal{N}(\mathcal{S}) \subset \mathcal{R}^3$ surrounding $\mathcal{S}$ (Marz & Macdonald, 2012) as an embedding in $\mathcal{R}^3$, which can be formally described by:

$$\mathcal{N}(\mathcal{S}) = \{x \in \mathcal{R} | \|x - cp(x)\| \le r\}, \tag{33}$$

where $r$ is called the tube-radius. The method is designed for solving PDE on the surface. The former definition enables us to formulate an embedding PDE on $\mathcal{N}(\mathcal{S})$, whose solution agrees with the solution of the surface PDE at the points $y \in \mathcal{S}$. More specifically, let $\tilde{u}(y)$ for $y \in \mathcal{S}$ and $u(x)$ for $x \in \mathcal{N}(\mathcal{S})$ denote the solution to the surface PDE and the embedding PDE, respectively. Fundamentally, the Closest Point Method (CPM) is based on extending surface from $\mathcal{S}$ onto $\mathcal{N}(\mathcal{S})$ such that the data is constant in the normal direction of $\mathcal{S}$. The task can be accomplished by the closest point extension, i.e. we take $u(x) = \tilde{u}(cp(x))$ for all $x \in \mathcal{N}(\mathcal{S})$. We can observe that the CP extension assigns surface data at the closest point of $x$ to $x$ itself. Then we solve the embedding PDE $u(x)$ and constrain the point on the surface to derive the solution. This extension also allows the differential forms on the surface to be replaced with the differential forms on the Cartesian differential forms (Ruuth & Merriman, 2008).

Hence, we can utilize the transform of differential forms on the surface to the differential forms on the neighborhood, parameterizing with it to preserve the certain property (divergence-free). Then we use the parametric functionals to solve $u$ of PDEs on the Cartesian neighborhood and pull it back to the surface with $u(x) = \tilde{u}(cp(x))$. Hence, our objective is to calculate the transformation of differential form that is required in PDE. The basic differential operator transformations such as exterior derivative, hodge star, wedge product $d, *, \wedge$ are showed in Li et al. (2023a) and Tab. 2. We next employ them to calculate the differential operator that we need to construct the corresponding divergence free field.

## B Proofs and derivations

### B.1 Proof for Theorem 3.1

We first propose the exterior calculus rules on 3D in Tab. 2, where $f_{(k)}$ means the $k$-form of $f$. $f$ and $g$ represent the scalar function and $\mathbf{u}, \mathbf{v}, \mathbf{w}$ represent vector functions.

| Output type | Wedge product $\wedge$ | Interior product $i_{\mathbf{u}}$ | Exterior derivative $d$ | Hodge star $*$ |
|---|---|---|---|---|
| 0-form | $f_{(0)} \wedge g_{(0)} = (fg)_{(0)}$ | $i_{\mathbf{u}}\mathbf{w}_{(1)} = (\mathbf{u} \cdot \mathbf{w})_{(0)}$ | N/A | $*f_{(3)} = f_{(0)}$ |
| 1-form | $f_{(0)} \wedge \mathbf{u}_{(1)} = f\mathbf{u}_{(1)}$ | $i_{\mathbf{u}}\mathbf{w}_{(2)} = (\mathbf{w} \times \mathbf{u})_{(1)}$ | $df_{(0)} = (\nabla f)_{(1)}$ | $*\mathbf{u}_{(2)} = \mathbf{u}_{(1)}$ |
| 2-form | $\mathbf{u}_{(1)} \wedge \mathbf{v}_{(1)} = (\mathbf{u} \times \mathbf{v})_{(2)}$ | $i_{\mathbf{u}}f_{(3)} = f\mathbf{u}_{(2)}$ | $d\mathbf{u}_{(1)} = (\nabla \times \mathbf{u})_{(2)}$ | $*\mathbf{u}_{(1)} = \mathbf{u}_{(2)}$ |
| 3-form | $\mathbf{u}_{(1)} \wedge \mathbf{v}_{(2)} = (\mathbf{u} \cdot \mathbf{v})_{(3)}$ | N/A | $d\mathbf{u}_{(2)} = (\nabla \cdot \mathbf{u})_{(3)}$ | $*f_{(0)} = f_{(3)}$ |

Table 2: Exterior calculus operators in 3D.

Based on Li et al. (2023a), we can further write down the exterior calculus with the Closest Point Method via $cp^*$. It allows us to emulate the operators on $\mathcal{S}$ (denoted with a superscript $\mathcal{S}$) using the operators on $\mathcal{R}^3$ (denoted with a superscript $\mathcal{R}^3$):

$$\text{CP-wedge product:} \quad cp^*(\alpha \wedge^{\mathcal{S}} \beta) = (cp^*\alpha) \wedge^{\mathcal{R}^3} (cp^*\beta), \tag{34a}$$

$$\text{CP-interior product:} \quad cp^*(i_{\mathbf{Fu}}^{\mathcal{S}}\alpha) = i_{\mathbf{u}}^{\mathcal{R}^3} cp^*\alpha, \tag{34b}$$

$$\text{CP-exterior derivative:} \quad cp^*(d^{\mathcal{S}}\alpha) = d^{\mathcal{R}^3} cp^*\alpha, \tag{34c}$$

$$\text{CP-hodge star:} \quad cp^*(*^{\mathcal{S}}\alpha)|_{j(\mathcal{S})} = i_{\boldsymbol{n}} *^{\mathcal{R}^3} (cp^*\alpha)|_{j(\mathcal{S})}, \tag{34d}$$

where $\mathbf{F} = d(j \circ cp)$ denotes the Jacobian and CP-hodge star is only applicable directly at the surface.

**Proof for Theorem 3.1:** We prove our theorem 3.1 by mathematical manipulations. As we state in Sec. 3.1, we need to construct $\boldsymbol{v} = *^{\mathcal{S}} d^{\mathcal{S}} \sigma$ via $\sigma$ on the surface. Then we apply $cp^*$ on both side, we derive

$$
\begin{aligned}
cp^* \boldsymbol{v} &= cp^*(*^{\mathcal{S}} d^{\mathcal{S}} \sigma) \\
&= i_{\boldsymbol{n}} *^{\mathcal{R}^3} (cp^* d^{\mathcal{S}} \sigma) \\
&= i_{\boldsymbol{n}} *^{\mathcal{R}^3} d^{\mathcal{R}^3} cp^* \sigma \\
&= i_{\boldsymbol{n}} *^{\mathcal{R}^3} \nabla(cp^* \sigma)_{(1)} \\
&= i_{\boldsymbol{n}} \nabla(cp^* \sigma)_{(2)} \\
&= \nabla(cp^* \sigma) \times \boldsymbol{n}.
\end{aligned}
\tag{35}
$$

Then on the surface, we have $(cp^* \boldsymbol{v})(j(x)) = \boldsymbol{v}(cp \circ j(x)) = \boldsymbol{v}(x)$ (the closest point for the surface point is itself). Hence, we can derive

$$
\boldsymbol{v} = (\nabla(cp^* \sigma) \circ j(x)) \times \boldsymbol{n}.
\tag{36}
$$

Then for the vorticity function in 2D, it is defined by $w = \operatorname{curl} \boldsymbol{v}$ as a scalar. For the surface, we derive the vorticity function via the hodge decomposition. We need to derive the divergence-free component of $\boldsymbol{v}$ (as "curl" on surfaces), which actually is $*d\boldsymbol{v}$. Then we can derive

$$
\begin{aligned}
cp^* \omega &= cp^*(*^{\mathcal{S}} d^{\mathcal{S}} \boldsymbol{v}) \\
&= i_{\boldsymbol{n}} *^{\mathcal{R}^3} (cp^* d^{\mathcal{S}} \boldsymbol{v}) \\
&= i_{\boldsymbol{n}} *^{\mathcal{R}^3} d^{\mathcal{R}^3} (cp^* \boldsymbol{v})_{(1)} \\
&= i_{\boldsymbol{n}} *^{\mathcal{R}^3} (\nabla \times (cp^* \boldsymbol{v}))_{(2)} \\
&= i_{\boldsymbol{n}} *^{\mathcal{R}^3} (\nabla \times (cp^* \boldsymbol{v}))_{(1)} \\
&= \nabla(cp^* \boldsymbol{v}) \cdot \boldsymbol{n}.
\end{aligned}
\tag{37}
$$

Similar with Eq. 35, we achieve

$$
\omega = (\nabla \times \boldsymbol{v}) \cdot \boldsymbol{n}.
\tag{38}
$$

$\square$

Note that on the surface, the rules work like ones in 2D: the stream and the vorticity functions are both scalars (compared with the vector potential functions in $\mathcal{R}^3$ (Elcott et al., 2007a; Chang et al., 2022)). It can also be verified with the exterior calculus. If $\sigma$ is not 0-form but 1-form instead, we can not derive $\boldsymbol{v}$ as 1-form but 0-form, which brings trouble for our neural representation.

**Remark:** We can next parameterize the $\sigma$ (with NN) and compute $\boldsymbol{v}$ and $\omega$ by sampling points and do calculus with Eqs. 36 and 38. The theorem provides a continuous and analytic formulation on the vorticity function on the surfaces with the help of the Closest Point Method in Tab. 2 while the classic method (Azencot et al., 2014) needs to compute the discretized differential form with the triangle meshes. CPM enables us to parameterize the surface field on the ambient space and facilitate the neural representation.

## C  CONDITIONING PROPERTY OF OUR FRAMEWORK

For the generation problems, to verify the capability, we mirror the architecture of variation auto-encoder. Let's consider images as an example input, and our objective is to provide a model that receive the coordinate $x$ and image conditions as an input and generate a divergence-free field of which vorticity $\omega$ resembles the corresponding conditioned images silhouette in visualization. Images $q$ are encoded by the parametric encoder with $\theta_q$ and the features are provided with the reparameterization trick $z_q(\theta_q, q) \sim \mathcal{N}(\xi(\theta_q, q), \tau(\theta_q, q))$ (Kingma & Welling, 2013), where $\xi(\theta_q, q), \tau(\theta_q, q)$ is the mean and variance output by the encoder. The encoded feature $z = (z_q, z_{aux})$, where $z_{aux}$ is auxiliary information such as the image class, is input together with positional embedding (like

siren) from $x$. Then the decoder translates the feature to $\sigma$ function and further derive $\omega$ with Theorem 3.1. The loss function consists of two parts: One is $\|\omega(\sigma(\theta, z(\theta_q))) - G(q)\|_2$, where $G$ is mapping from the image color space to vorticity space. This term supervises the vorticity field to shape as the silhouette of the corresponding images. The other term is Kullback-Leibler divergence to constrain the distribution of $z_q$ to be normal, as given by $\|\xi^2 + \tau^2 - \log \tau^2 + 1\|_2$ with $\xi, \tau \in \mathcal{R}^r$, where $r$ is the dimension of $z_q$ and $\log$ operation is computed element-wise. With such design, we could derive a simple conditioning framework, which enables more data prior into the simulation via a more end-to-end manner. More details are also provided in Appendix G. Our neural parameterization provides more direct and effective approach to involve the semantic information and show more potential to combine with the advanced generation methods in 3D or language model.

**Remark:** The idea of the conditioning resonates with the eigen-decomposition of the fluid fields, as discussed in Cui et al. (2021). This method decomposes the vector field $\boldsymbol{v}$ into several divergence-free basis $\boldsymbol{u}_i$, i.e. $\boldsymbol{v} = \sum_{i=1}^{r} w_i \boldsymbol{u}_i$ and $\boldsymbol{u}_i$ is the eigenfunction of the Laplacian operator. It's noteworthy that $w_i$ can serve as the encoding $z$ in our formulation. If we strengthen our neural represented velocity field to become eigenfunctions of the Laplacian operator on surfaces, we can derive similar decomposition weights through the conditioning process described above.

## D   PSEUDOCODE FOR ADVECTION

---
**Algorithm 1** Advection for Neural Flow on Surfaces.
---
**Input:** Initial velocity field $\boldsymbol{v}_0$, vorticity field $\omega_0$, timestep size $h$, number of timesteps $N$, surface $\mathcal{S}$, training steps $E$, sample size $k$, learning rate $\alpha$
Fitting the initial network weight $\theta_0$ and non-zero harmonic term $\eta$ with $\boldsymbol{v}_0$ and $\omega_0$.
**for** $n = 1$ **to** $N$ **do**
   $\theta_{n+1} \leftarrow \theta_n$
   **for** $i = 1$ **to** $E$ **do**
      Sample $k$ point on $\mathcal{S}$ as sample set $\mathcal{M}$.
      Compute the stream function $\sigma(\theta_{n+1})$ on $\mathcal{M}$.
      Compute the velocity $\boldsymbol{v}(\theta_n)$, $\boldsymbol{v}(\theta_{n+1})$ and vorticity $\omega(\theta_n)$, $\omega(\theta_{n+1})$ with Eqs. 4 and 5 for time $n$ and $n+1$ with $\sigma(\theta_n)$, $\sigma(\theta_{n+1})$ and $\eta$.
      Construct loss function $\mathcal{L}_{\theta_{n+1}}$ with Eq. 15 for $\mathcal{M}$.
      $\theta_{n+1} \leftarrow \theta_{n+1} - \alpha \nabla \mathcal{L}_{\theta_{n+1}}$
   **end for**
**end for**
---

## E   ADDITIONAL EXPERIMENTAL RESULTS

We include additional experiments to verify the performances of our method as a supplementary. We provide the quantitative study for the flow on the inclined plane, the ablation and comparison on the sphere jet flow and the convergence study for our flow on the implicit neural representation.

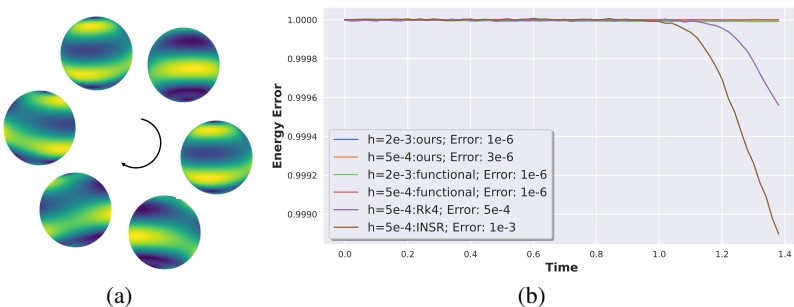

| (a) | (b) |

Figure 9: Results for rotating sphere flow. (a) Qualitative results for rotating sphere. (b) Quantitative results for energy preservation.

### E.1 RESULTS FOR ROTATING SPHERE FLOW

We verify the energy conservation property of our method by examining the rotating sphere flow, using an analytic solution as validated by Azencot et al. (2014). This approach allows us to rigorously test our simulator's reliability in preserving energy over time. The initial flow conditions combine a Killing vector field with a rotated gradient of an eigenfunction of the Laplace-Beltrami operator on the surface. Killing vector filed is a vector field whose Lie derivative of the metric vanishes, meaning that the flow generated by the vector field remains constant in advection (Jost & Jost, 2008). A classic example on the sphere is the vector field $(-y, x, 0)$ around $z$-axis. The eigenfunctions of the Laplace-Beltrami operator on the sphere are well-known as Spherical Harmonics functions. We also apply the rotation on them for better identification. Actually, this rotating sphere flow appears as if the sphere with the vector field is rotating over time when observed from a fixed point. It can be demonstrated that the energy of inviscid flow with these initial conditions remains constant, making this configuration an excellent test case for energy conservation. As the results showed in Fig. 9 (a), the results exhibit periodic patterns resembling the sphere rotating.

We also plot the energy change across the entire sphere, comparing it with Functional Fluid on Surfaces (Azencot et al., 2014), the classic method with Runge-Kutta (RK) time integrator and Implicit Neural Spatial Representations (INSR) (Chen et al., 2023) with semi-Lagrangian advection as the loss function. The results indicate that our method achieves a relative change in energy on the order of $10^{-5}$, which is comparable to the results from the Functional Fluid on Surfaces method for both larger and smaller time steps. Conversely, the RK method and INSR suffers from larger energy losses, even for smaller time steps.

### E.2 MORE RESULTS FOR THE TAYLOR VORTICES ON THE INCLINED PLANE

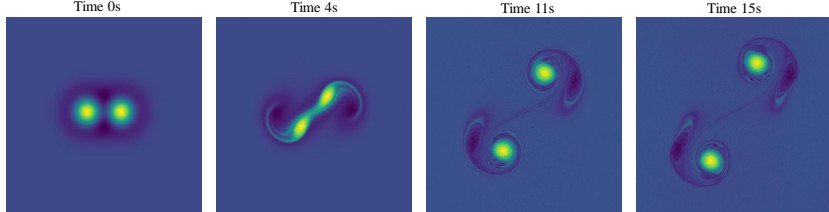

Figure 10: Dynamic Results for Taylor vortices on inclined plane.

We include the qualitative results of our simulation dynamics for Taylor vortices in Fig. 10 and more comparison results are provided in the supplementary video. We can observe a clear vortex pair and the phenomenon of separation.

| Methods | Error | Time | Storage |
|---------|-------|------|---------|
| PINN | 4.16e5 | 9.6 h | 568.1KB |
| INSR | 3.45e3 | 16.8 h | **516.3KB** |
| Small-F.S. | 3.21e2 | **0.2h** | 535.6KB |
| HomoLBM | 8.92e1 | 3.8 h | 4.3MB |
| Small-F.S. | 3.21e2 | **0.2h** | 535.6KB |
| Ours | **1.71e1** | 12.7 h | 521.3KB |
| GT | N/A | 2.3 h | 3964.0KB |

Table 3: Quantitative results for the Taylor vortices on inclined plane. Error: mean square error (MSE) averaged by 120 time steps with resolution of 400. The storage of our high resolution ground truth is 7.6 times than ours.

We also include the quantitative results for the Taylor vortices in Tab. 3, together with more comparison with classic methods on the 2D plane for benchmark studies, including Stable Fluid (with a resolution of 1024x1024) (Stam, 1999) and high-order Lattice Boltzmann methods (not parallel version with a resolution of 512x512) (Li et al., 2023b). The corresponding qualitative results are

presented in Fig. 11. The results demonstrate that our method enables high accuracy while showing high memory-efficiency, compared with both classical and recent advanced methods under the similar memory cost.

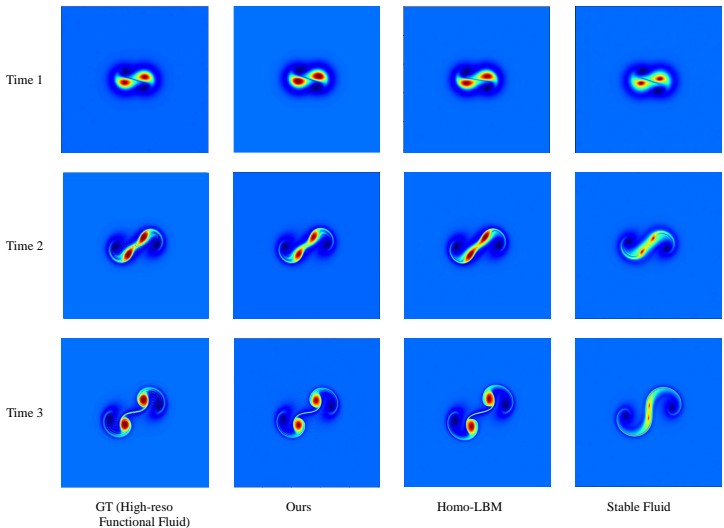

Figure 11: Qualitative results for the Taylor vortices compared with 2D classical methods.

### E.3 MORE COMPARISON RESULTS ON THE SPHERE JET FLOW

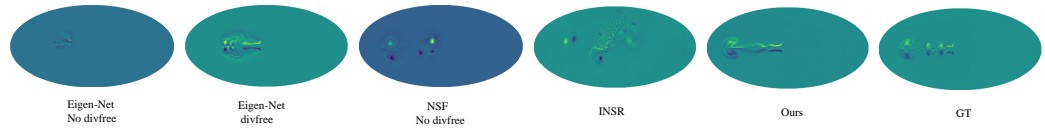

Figure 12: Qualitative results for the sphere jet flow compared with eigen-net and NSF.

We compare with other surface field representation methods to verify the effectiveness of our proposed framework. We try to adapt two methods into our framework. One is Koestler et al. (Koestler et al., 2022), which proposes a surface field representation via the eigenfunction of the Laplacian-beltrami operator (eigen-net) on the surface and the other is Xue et al. (Xue et al., 2023) that utilizes the MLP and projection operator (NSF). We implement them on the sphere jet flow case for comparison since the Laplacian-beltrami operator can be analytically computed by Spherical Harmonics. To further verify the effectiveness of our field function design, for eigen-net, we implemented two variations as ablations: one that incorporates our divergence-free design along with the covariant derivative advection, and another that avoids our divergence-free approach in favor of the traditional advection and divergence projection; for NSF we adopted the method without our divergence-free design, as incorporating it would result in a configuration similar to our own framework, with the primary distinction being the hyper-parameters of the MLP or siren. We show the quantitative results in the Tab. 4 and the corresponding qualitative results in Fig. 12.

Both quantitative and qualitative results demonstrate that our proposed framework achieves higher accuracy and memory-efficiency in the incompressible Euler flow simulation, compared with pure eigen-net or NSF surface field representations. The ablation results also indicate that that our proposed divergence-free field design framework significantly improves the accuracy of the eigen-net. Ours avoid the divergence-free projection, which reduces the extra fitting error and cascading error effects. The results also illustrate that our proposed divergence-free design is robust with the ways of parameterization (MLP, siren or eigen-net).

| Methods | Error | Time | Storage |
|---|---|---|---|
| Eigen-net without divergence-free design | 6.82e6 | 13.5 h | 632.5KB |
| NSF without divergence-free design | 9.76e4 | **12.8 h** | **501.3 KB** |
| Eigen-net with divergence-free | 1.13e3 | 16.7h | 622.4KB |
| Ours | **2.89e2** | 16.5 h | 532.8KB |

Table 4: Quantitative results for the sphere jet flow compared with eigen-net and NSF. Error: mean square error (MSE) averaged by 100 time steps on 81924 mesh vertices.

| Methods | Error | Time | Storage |
|---|---|---|---|
| Elcott et al. 2007 | 1.27e4 | 6.8h | 2643.0KB |
| Stable Fluid | 8.62e5 | **0.2h** | **972.8KB** |
| Small-F.S. | 5.34e3 | 0.8h | 583.8KB |
| Ours | **2.89e2** | 16.5 h | 532.8KB |
| GT | N/A | 8.3 h | 2643.0 KB |

Table 5: More Quantitative results for the sphere jet flow with classic methods.

We also include more results of classical solvers for reference, including Stable Fluid (with a resolution of 256x256) (Stam, 1999) and Elcott et al. (Elcott et al., 2007a) (using the same mesh as GT). The quantitative results are shown in Tab. 5, and the qualitative results are presented in Fig. 13.

The results show that our method achieves high performance and low energy dissipation compared to classic methods. However, this improvement comes with the trade-off of higher time consumption as discuss in Appendix F.3.

### E.4 CONVERGENCE VERIFICATION FOR THE FLOW ON THE IMPLICIT REPRESENTED SURFACES

| Marching Cubes resolution | Average steps of Crashing | Storage |
|---|---|---|
| 64 | 36.5 | 117.18 KB |
| 128 | 53.4 | 768.32 KB |
| 240 | 68.7 | 1123.15 KB |
| GT Mesh | 88.6 | 1419.48 KB |
| Ours | N/A | **523.4 KB** |

Table 6: Time steps that the classic method (Azencot et al., 2014) with Marching Cubes on implicit neural represented surfaces crashes. We repeat the process with 10 times and derive the averaged results. However, ours can work well with the implicit neural representation.

For the case of implicit represented surfaces, as previously stated in the main context (Sec. 5.2), for the classical method it is not convergent, leading to non-referable results. To substantiate our claim regarding non-convergence, we include a new Tab. 6 that lists the simulation crashing time steps across different marching cube resolutions for the traditional methods. This data will offer a quantitative perspective on the limitations of the classic methods in robustness. We also include the reference qualitative results in the supplementary video to exhibit the simulation process and describe the non convergence on different resolutions. The results demonstrate that our meshless and end-to-end method can achieve high adaptability and robustness for the simulation on the implicit neural representation, while the classical method fails in marching cube meshes and needs further complex geometry processing schemes to improve the mesh quality.

### E.5 ABLATION STUDIES FOR THE NETWORK AND TRAINING DESIGN

We also include the ablation studies on the network size and sample count on our sphere jet case. The results are exhibited in Tab. 7 and Tab. 8, which indicate that our method is relatively insensitive to these settings unless the width is extremely small.

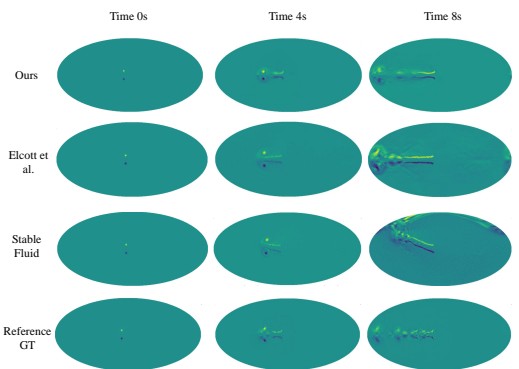

Figure 13: Qualitative results for the sphere jet flow compared with more classical methods.

| Network Size | Error |
|---|---|
| 4 layers, 128 | 2.89e2 |
| 4 layers, 256 | 2.21e2 |
| 4 layers, 64 | 4.57e2 |

Table 7: Ablation studies for network size for the sphere jet flow.

## F DISCUSSIONS AND LIMITATIONS

### F.1 DISCUSSIONS AND LIMITATIONS ON THE TOPOLOGY PROBLEMS

The first limitation is about the singularity problem. We don't focus heavily on the singularity issues related to topology, such as those highlighted by the Poincaré-Hopf Theorem, which is a complex challenge. Our main interest lies in using neural approaches to simulate fluid dynamics and model vortex dynamics on surfaces for visual effects (as described in Eq. 6), rather than dealing with arbitrary vector fields. The scenes chosen in both Azencot et al. (2014); Ando et al. (2015) (which do not mention the issue) and ours typically maintain non-zero measure zero velocity and avoid poles to simplify the analysis. We initialize the simulation with finite vorticity, which makes it hard for advection to generate infinite vorticity (or singularities) within our settings. Furthermore, we employ the stream and curl regularization to avoid extreme values in the singularity points and preserve system stability. According to Sard's theorem, the number of poles has zero measure in our simple cases, we maintain a very small probability of sampling at the poles, which improves the stability of the simulation. Though the velocity field near singularities will be underfitted and approximated by the network, potentially leading to inaccuracies and numerical dissipation near the "cyclone", the overall results provide empirically reasonable visual effects, which is the primary goal of our application.

To further validate the effectiveness of our regularization and sampling, we applied our framework to fit the velocity $(-\sin(\theta), \cos(\theta))$ on a sphere with spherical coordinates $(1, \theta, \phi)$, which keeps a singularity at the northern/southern pole $((1, 0, 0), (1, \pi, 0))$. The errors and velocity magnitude are shown in Fig. 14, where we achieve the desired velocity pattern. Although some errors appear in the initial epochs, they become negligible over time, which is sufficient for graphics and visual effects.

| Training Samples (millions) | Error |
|---|---|
| 60 | 2.89e2 |
| 40 | 3.12e2 |
| 80 | 2.79e2 |

Table 8: Ablation studies for sample number for the sphere jet flow.

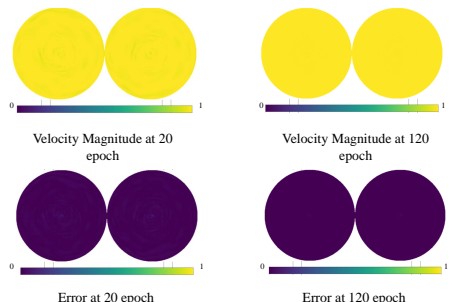

Figure 14: Qualitative results of the numerical studies (Velocity magnitude and error) for the constant sphere flow. (Views from the northern and southern poles)

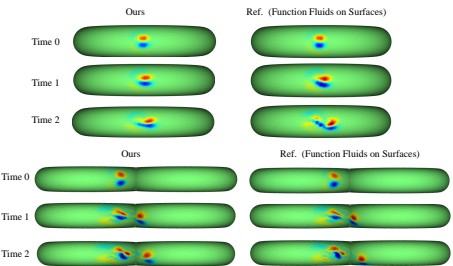

Figure 15: Qualitative results for the sphere jet flow on the explicit torus and double torus.

Another problem related with the topology is the cohomology term. Our method is mainly adopted for the topology for genius zero, where the velocity can be estimated by the stream function without more consideration of the time-variant cohomology component (as stated in Yin et al. (2023) Proposition 5 for the surfaces removing several poles). For the surface with high genius, we find it insufficient for the time-invariant cohomology term to handle highly turbulent flow and generate complex visual effects. The problem can be further addressed by incorporating the solver in Yin et al. (2023) with the neural network, which will be our future work. However, only for the visual effects, our method still performs well for the high genius surfaces and shows the correct jet flow behavior compared to the reference classic method. We plot the results in Fig. 15 with simple torus and double torus cases with explicit torus and double torus meshes. They do demonstrate reasonable visual effects on high-genus surfaces even without considering the time-variant cohomology.

## F.2 DISCUSSIONS AND LIMITATIONS ON THE GEOMETRY PROBLEMS

As stated in Sec. 3.2, our method needs SDF for the flow on the implicit surfaces. This can pose challenges when calculating normals for open surfaces. To overcome this, one can use an unsigned distance field (UDF) to extend our framework to open surfaces, which is well applied in Chibane et al. (2020); Yang et al. (2023); Long et al. (2023).

Additionally, our framework is based on the Closest Point Method (CPM), which, due to its reliance on ambient space, faces challenges when dealing with narrow geometric features. This reliance can lead to ambiguities when processing narrow or thin features, resulting in inefficiency and inaccuracy due to increased difficulties in convergence. To mitigate this issue, we can follow the approaches in King et al. (2023); Marz & Macdonald (2012) and sample the neighborhood within a small tube radius adaptively. In parctice, the sampling distance threshold to the surface can be determined following the method outlined in King et al. (2023) (Sec. 8), guided by the estimation of the surface reach distance as described by Aamari et al. (2019). This estimation can be locally constructed using a simple mesh extractor from the signed distance function (SDF) if no mesh is available. This allows for rough detection of thin regions and adaptive adjustment of the sampling distance, thereby improving both efficiency and accuracy.

Moreover, in the practical scenarios, the non-smooth surface, imperfect SDF (common problems in implicit neural representations such as Yifan et al. (2021)) and the complex surfaces (like unoriented surfaces) can also limit our performances. Noisy surfaces and normals introduce numerical viscosity, slowing down the simulation. But fortunately, we show that they do not drive our simulation to crash, as the neural representation provides smoother and more robust results (as demonstrated in Appendix E.4 with GT mesh data) compared to traditional methods. To further solve these issues, we can construct smooth approximator and utilize a large number of samples from a smaller tube radius (as described in King et al. (2023) above) for improved performance. Additionally, the orientation problem is addressed in King et al. (2023) and the extension for our method c an be similarly constructed.

### F.3 DISCUSSIONS AND LIMITATIONS ON THE TIME AND MEMORY CONSUMPTION

Our method actually keeps large advantages at the memory consumption albeit with the increased time requirements. The low memory usage is crucial for handling and analyzing high-resolution data. For example, a 2D simulation at a 4096x4096 resolution requires about 200MB per frame, resulting in considerable memory demands for extended simulations. Similarly, high-resolution 3D grid simulations consume even more memory. Although mesh-based simulations can be more efficient, they encounter mesh quality issues, as shown in the robustness tests in Appendix E.4. The time cost remains high, as methods like siren or simple MLPs are too global and inefficient for optimization, and inference sampling (particularly for implicit neural representations) is time-consuming. To address the time consumption issue, a hybrid simulator to achieve both high speed and performances is necessary. Fortunately, recent advances in hybrid representations (Müller et al., 2022; Huang et al., 2023) have shown promising results in reducing training time from hours to seconds while maintaining the expressiveness. Employing these more efficient representations hold great promise for improvements of our method. In the inference time, a sampler (Sharp & Jacobson, 2022) with high efficiency can be adopted, which will provide huge saving for the inference time with KD hierarchies.

## G   MORE IMPLEMENTATION DETAILS

We provide our implementation details for our numerical studies. Our experiments are all implemented with Jax library (Bradbury et al., 2018) on an NVIDIA GeForce RTX 3090 GPU.

**Sphere Jet:** We adopt the 4-layers MLP (for shaper simulation results compared with siren) with 128 units for our implementation. The learning rate is set with the exponential decay from $1e-5$ to $1e-7$ with 60000 steps and batch size 1000 for each time step. The time step is chosen as $5e-2$. The initialization is the same as Azencot et al. (2014) and the initial vorticity is kept for the whole simulation time. For the comparison, INSR uses the siren function with 4-layers 128 units for advection, projection and correction respectively. The learning rate is set as $1e-6$. For each process, the siren iterates for 40000 steps with batch size 5000. For the original PINN, we adopt the MLP with 4-layers 128 units. The loss function is to enforce the incompressible Euler equation directly, as showed in Appendix A.3 of Chen et al. (2023). The learning rate is set by $1e-5$ with 60000 steps and batch size 2000.

**Taylor vortices on inclined plane:** We adopt 4-layer siren with 128 units representation to conduct the positional encoding with the first layer frequency as 30. The time step is chosen as $5e-3$. The initialization is set the same as McKenzie (2007), and we rotate the plane and make the normal be $(0.3, -0.5, 0.8)$. The domain size is set $[-\pi, \pi]$ with the periodic boundary condition. The time step is set as 0.05. The learning rate is set with the piece-wise constant from $1e-5$ with a decay factor 0.1 on 40000 and 60000 steps for total 80000 steps and batch size 1000. For comparison, INSR adopts the siren MLP with 4 layers and 128 units per layer for advection, projection and correction respectively. The learning rate is set as $1e-5$. For each process, the siren iterates for 20000 steps with batch size 1000. For the original PINN, we adopt MLP with 4 layers and 128 units. The loss function consists of the governing equation part which is the same as the one in sphere jet case and the periodic boundary condition part. The learning rate is set by $1e-5$ with 60000 iterations and batch size 1000.

**Rotating sphere flow:** For the construction of $\sigma$, we adopt 4-layer siren with 128 units representation (Sitzmann et al., 2020) with the first layer frequency as 30. The learning rate is set with the exponential decay from $1e-6$ to $1e-8$ with 40000 steps and batch size 1000 for each time step. The time step is chosen as $5e-4$ and $2e-3$. The initialization of the velocity is set as the Killing field $(-y, x, 0)$ with the rotated 4-degree Spherical Harmonics functions with order 4 and 5. For INSR for our comparison, we adopt the siren function with 4-layers 64 units for advection, projection and correction respectively. The learning rate is set as $5e-4$. For each process, the siren iterates for 10000 steps with batch size 1000.

**Flow on the explicit meshes.** We adopt a 4-layer siren with 128 units representation for both models with the first layer frequency as 30. The learning rate is set with the exponential decay from $1e-6$ to $1e-8$ with 40000 steps and batch size 2400 for each time step. The time step is chosen as $5e-2$ and $8e-2$ for the hand and spot respectively. The hand model is initialized by two vortices with the geodesic 0.41 by Crane et al. (2013b) and the spot model is set with 0.3.

**Flow on the implicit mesh.** First, we adopt 4-layer siren with 256 units to reconstruct the implicit neural representation of the SDF function based on Sitzmann et al. (2020) with the uniform sampling. The in the simulation, we take the rejection rules in Yang et al. (2021) to complete the uniformly sampling for the simulation function and avoid the samples concentrating near the high curvature area. The jet vortices are initialized by given two points on the mesh as an opposite pair. Our simulation neural field is also implemented as 4-layer siren with 128 units representation with the first layer frequency as 30. The learning rate is set with the exponential decay from $1e-5$ to $1e-7$ with 40000 steps and batch size 1000 for each time step. The time step is set by $5e-2$ and $2e-2$ for Armadillo and Lucy respectively.

**Flow with conditioning:** We adopt the encoder consists of 2-layer feature extraction MLP to reduce image data to a 32-dimension feature space; siren network to generate 128-dimensional feature with the input position $x$ as neural field representation and an one-hot class encoder for image categorization. Note that the first layer frequency for the siren network above is also 30. The decoder is a 2-layer MLP that transforms the concatenated features from the encoders to the stream function $\sigma$. Then following our Theorem 3.1, we can also derive the corresponding $v$ and $\omega$ taking the derivative with respect to $x$. In the inference time 32-dimension random Gaussian vector and the corresponding image class are input to concatenate with the positional encoding to generate the field value at the spatial point. The learning rate for the training process is set with the exponential decay from $1e-4$ to $1e-5$ with 400000 steps and batch size 1000. We map the color of the input images to the vorticity function and try to make the velocity field preserving the vortices shape as alphabets via Mean Least Square loss and KL loss.

**Flow for Helmholtz decomposition:** We adopt the atmosphere data (100-metre wind velocity) on Jan, 2024 (Raoult et al., 2017). We adopt 4-layer siren with 128 units representation for both models with the first layer frequency as 100. The learning rate is set with the exponential decay from $1e-4$ to $1e-5$ with 200000 steps and batch size 1000. We try to make our velocity results closer to the given data and derive the divergence-free component, similar as Richter-Powell et al. (2022).

