# OpenReview forum: "Neural Fluid Simulation on Geometric Surfaces"
_ICLR.cc/2025/Conference — ICLR 2025 Poster_

### Official Review · Reviewer_V8TK · 2024-10-29

**Soundness:** 1
**Presentation:** 1
**Contribution:** 1
**Rating:** 1
**Confidence:** 5

**Summary:**

This paper proposes an implicit neural representation to improve solvers that simulate flows on geometric surfaces through geometric adaptivity. The authors propose a neural physical simulation framework to construct a parameterized vector field on surfaces using exterior calculus formalism. Through a Closest Point Method, it is proposed an implicit neural network representation that is able to maintain a divergence-free property intrinsically. Divergence-free is an important property of Navier-Stokes solvers, and strictly enforcing them is a challenging task. Furthermore, the authors claim that the proposed approach is able to accurately preserve the energy of the flow as time advances.

**Strengths:**

Proposing alternative representations to the standard discretizations (e.g., grid/meshes) for solving PDEs is a very interesting and challenging topic of research. The authors propose a method that considers specific intricacies of the PDE solution when employing neural representations, along with desired properties that can potentially be satisfied in a continuous fashion (e.g., the divergence-free condition).

**Weaknesses:**

Unfortunately this paper is clearly below the ICLR quality acceptance bar. My major concerns are as follows:
- Poor exposition along with several typos which make the paper hard to understand. For example. the structure of Section 3.1 is composed of fragmented phrases forming very short paragraphs, making it hard to follow. Several typos and confusing phrasal structures (L11: “Incompressible Euler fluid on the surface”, L20: “We contribute a neural physical simulation framework on the surface with the implicit neural”, L240: “In the meanwhile" to name a few) greatly compromise the quality of the paper.
- The main idea of the paper is based on wrong assumptions. The poster “Closest Point Exterior Calculus”, in which the paper is heavily based, already offers a solution that is independent of the mesh quality. This invalidates one of the main motivations of this submission that previous approaches are dependent on mesh quality, and thus an implicit neural representation is required. Moreover, the assumption that storage is a limiting factor on solvers is also incorrect, since a solver usually has to store a single time-step of the represented variables for advancing the simulation state. The presented results also show very modest resolutions.
- There are missing references and/or previous methods are not thoroughly considered, leading to an outdated methodology proposition. Recent approaches (“Covector Fluids”, “Impulse Particle In Cell”, “Fluid Simulation on Neural Flow Maps”, “Eulerian-Lagrangian Fluid Simulation on Particle Flow Maps” and “Lagrangian Covector Fluid with Free Surface” to name a few) adopt structure preserving integrators by considering the deformation of the flow map during advection. This is ignored by the proposed advection method, which has a rather lengthy description in the paper. Lastly, Elcott et al, 2007b does not suffer from instabilities as it is mentioned in the manuscript and modern structure preserving solvers ("Impulse Particle In Cell”) are able to accurately advect velocities without major stability issues.
- The paper partially focuses on showing mathematical proofs that are known by the exterior calculus community (divergence free vector fields on surfaces), which make the described theory not so relevant as new theoretical contributions. The authors could just reference relevant discrete exterior calculus material or move the lengthy mathematical descriptions to the Appendix.
- The paper should have been focused on more relevant aspects of the implicit neural representation, such as network structure, how to properly tackle high-frequencies of the implicit neural field, how to make the training/evaluation process efficient (e.g., check “Instant Neural Graphics Primitives with a Multiresolution Hash Encoding”), etc.
- The authors mention that pressure projection (usually the most expensive part of a fluid solver) is not required by their approach. However, they solve a non-linear optimization problem iteratively with a simple ADAM gradient descent approach. This approach is way less efficient than traditional operator splitting, as evidenced by the timings shown in Table 1 (16h for 80k vertices is a very inefficient timing for the considered resolution). Lastly, there seems to be some high-frequency “ringing” artifacts generated by the proposed method in Figure 3 which are not present in ground truth or in the HOLA-7 results.

These are some of the reasons that justify my low score for this paper. I suggest the authors to rethink their approach before resubmitting the manuscript.

**Questions:**

- Did the authors explore alternative network designs for representing the implicit neural fields (such as "Instant Neural Graphics Primitives with a Multiresolution Hash Encoding”) ?
- How does the method fares in simulations where regions of turbulence are highly concentrated? Is the proposed adaptivity property working as expected?

---

> ### Author Response · Authors · 2024-11-17
>
> We thank the reviewer for the suggestion. To addreses your questions and concerns:
>
>
> > **Q1:** Poor exposition along with several typos.
>
> **R1:** For any writing issues are present, we assure that we will conduct a thorough proof reading and optimize the structure and descriptions throughout the paper. (L11: revised ``Incompressible Euler fluid`` to ``Incompressible fluid``;
> L20: corrected to ``We propose a neural physical simulation framework on the surface with the implicit neural representation``; L240 and others: rephased to improve transitions and ensure smoother readability).
>
> > **Q2:** The main idea of the paper is based on wrong assumptions ... since a solver usually has to store a single time-step of the represented variables for advancing the simulation state. The presented results also show very modest resolutions.
>
> **R2:**
> 1. Motivation Clarification. To clarify, we do not attribute the independence from mesh quality solely to the implicit neural
> representation. In fact, there exist many methods that do not incorporate the Closest Point Method face challenges related to mesh quality, and our goal is to improve these approaches. Even when using the Closest Point Method with a discretized grid, discretization errors can still occur in operator calculations. By leveraging neural parameterization, we achieve analytically accurate operator computation, which enhances robustness and addresses these challenges more effectively.
>
>  2. Storage Problem. Regarding the storage concern, we acknowledge that one-step computation is often sufficient for fast local result previews. However, for studies involving long-term effects or rendering on arbitrary paths (such as in scientific research or visual effects), storage becomes a significant consideration. Additionally, for scenarios involving data sharing or long-term analysis, storage efficiency cannot be overlooked. Our model addresses these challenges by supporting continuous input and enabling to produce very high-resolution results. For comparison in the content, we believe on the current resolution is adequate to show the effectiveness and robustness of our results while higher resolution simulation for the classic method will suffer from storage and robustness problem.
>
> > **Q3:** There are missing references and/or previous methods are not thoroughly considered ...  modern structure preserving solver accurately advect velocities without major stability issues.
>
> **R3:**  The paper referred here is mainly designed to deal with the numerical dissaption caused by the adveciton and projection. But none of them are designed for the surface flow (even not with a surface flow demo). While their methods could theoretically be extended to surface flows, the additional operators they propose would require careful adaptation and design for surface-specific contexts, which is beyond the scope of this work. To be more specific, Covector Fluid is relatively suitable for surface flows, but it still needs to specify and compute the co-vector form directly on the surface. Methods like Fluid Simulation on Neural Flow Maps, Eulerian-Lagrangian Fluid Simulation on Particle Flow Maps, and Impulse Particle In Cell rely on flow map forms, which could introduce possibility of instability due to surface operator estimation error when applied to surfaces. Lagrangian Covector Fluid with Free Surface differs slightly due to its use of power graph discretization, which also presents extra challenges for extension to surfaces.
>
> Regarding Elcott et al. 2007 [5], the computation of flow lines on triangle meshes is known to introduce numerical instability, as discussed in Anzenot et al. 2014 [6]. Although Impulse Particle In Cell shows potential to solve the problem on the surface but still needs further verfication, which is beyond our scope of the paper.
>
> > **Q4:** The paper partially focuses on showing mathematical proofs that are known by the exterior calculus community ... move the lengthy mathematical descriptions to the Appendix.
>
> **R4:**  We believe this is not a direct consequence of solely the divergence-free field construction using exterior calculus and discrete differential geometry as cited. Our approach requires the use of an additional tool based on the Closest Point Method (CPM), as described in [7], to theoretically identify the stream function and other necessary differential forms. Our theoretical formulation provides a more rigorous construction of the "surface curl" and enables the application of neural parametric functions for surface vector field dynamics. To assist readers who may not be familiar with these techniques, we aim to include a comprehensive sketch in the appendix. We believe this addition will provide valuable insights and improve the accessibility of our methodology.

---

> > ### Author Response · Authors · 2024-11-17
> >
> > > **Q5:** The paper should have been focused on more relevant aspects of the implicit neural representation, such as network structure, how to properly tackle high-frequencies of the implicit neural field,  how to make the training/evaluation process efficient (e.g., check “Instant Neural Graphics Primitives with a Multiresolution Hash Encoding”).
> >
> > **R5:** Our work primarily focuses on the neural parameterization of surface divergence-free vector fields. The first step is to model the field using a neural network, after which we can explore further optimizations, such as improving network design, enhancing structure, or increasing efficiency using other implicit neural representation (INR) architectures (e.g., Neural Graphics Primitives, NGP). These directions are part of our future work and are discussed in detail in Appendix F.3.
> >
> >
> > > **Q6:** The authors mention that pressure projection (usually the most expensive part of a fluid solver) is not required by their approach. ... Lastly, there seems to be some high-frequency “ringing” artifacts generated by the proposed method in Figure 3 which are not present in ground truth or in the HOLA-7 results.
> >
> > **R6:** We acknowledge the limitation of our method in terms of time consumption, which has been explicitly discussed in both Sec. 6 and Appendix F.3. In contrast to Chorin's operator splitting scheme, our approach guarantees divergence-free behavior in a functional manner. In comparison, the advection-projection method introduces numerical diffusion, as noted in prior works (Chang et al., 2002 [8]; Elcott et al., 2007 [5]). While we employ a first-order optimizer for the non-linear problem, which may result in numerical inaccuracies, our method is still a better alternative for the classic ones both theoretically (as a form of Lie advection for divergence free field) and empirically (in the energy presevation studies). Regarding the "ringing effect," we are uncertain whether it occurs in our method. For implicit neural surface representations (INSR), it might arise due to the Siren parameterization, and certain periodic patterns are exhibited.
> >
> >
> > > **Q7:** Did the authors explore alternative network designs for representing the implicit neural fields (such as "Instant Neural Graphics Primitives with a Multiresolution Hash Encoding”) ?
> >
> > **R7:**  In our work, we focused on using MLP and Siren for analytic operator computation. The use of Neural Graphics Primitives (NGP) is mentioned in Appendix F.3 as a potential future direction to improve efficiency. However, incorporating higher-order differential operators into NGP requires more careful design, as highlighted in Li et al. [9].
> >
> >
> > > **Q8:** How does the method fares in simulations where regions of turbulence are highly concentrated? Is the proposed adaptivity property working as expected?
> >
> > **R8:** Our method may introduce some numerical dissipation in highly turbulent regions due to limitations in model representation (e.g., maximum frequency in Siren) and optimization. This can result in errors that reduce the turbulence intensity compared to the desired level. The adaptivity discussed in our paper primarily refers to the ability to handle different geometry representations, which has been evaluated through numerical studies using analytic, mesh, and INR cases. However, turbulence is influenced not only by geometry but also by the characteristics of the vector field itself. We believe that the adaptivity property is not directly related to turbulent simulations.

---

> > > ### Author Response · Authors · 2024-11-17
> > >
> > > **Reference:**
> > >
> > > [5]. Elcott S, Tong Y, Kanso E, et al. Stable, circulation-preserving, simplicial fluids[J]. ACM Transactions on Graphics (TOG), 2007, 26(1): 4-es.
> > >
> > > [6]. Azencot O, Weißmann S, Ovsjanikov M, et al. Functional fluids on surfaces[C]//Computer Graphics Forum. 2014, 33(5): 237-246.
> > >
> > > [7]. Li M, Owens M, Wu J, et al. Closest Point Exterior Calculus[M]//SIGGRAPH Asia 2023 Posters. 2023: 1-2.
> > >
> > > [8]. Chang W, Giraldo F, Perot B. Analysis of an exact fractional step method[J]. Journal of Computational Physics, 2002, 180(1): 183-199.
> > >
> > > [9]. Li Z, Müller T, Evans A, et al. Neuralangelo: High-fidelity neural surface reconstruction[C]//Proceedings of the IEEE/CVF Conference on Computer Vision and Pattern Recognition. 2023: 8456-8465.

---

> ### Comment · Reviewer_V8TK · 2024-11-22
>
> Honestly I’m a bit surprised by the other reviews, specially the one that rated this submission a 10. I strongly believe that this manuscript, in its current form, is not ready for publication. I think this paper would benefit from another submission cycle, carefully reassessing the underlying assumptions about the method. I restate that my major concerns are related to **paper usefulness** (the proposed method is extremely slow (17 hours for a 2D mesh!!) and has little application), **exposition** (poor English, several typos and I just noticed a few on my original rebuttal), **lack of proper evaluation** (few examples, some of them with noticeable artifacts, quantitative evaluation is poor), **little contributions** (its widely known that taking the curl of the streamfunction yields a div-free velocity field, plugging CPM into a neural representation is not a major breakthrough) and several important **related works are missing**. Moreover, employing streamfunctions to solve flow equations difficult boundary conditions (not addressed at all in the paper), and it is a reason why the majority of recent approaches in Computer Graphics are using impulse-based formulations for energy preservation.
>
> Let me address a few inconsistencies on the rebuttal:
>
> _"To clarify, we do not attribute the independence from mesh quality solely to the implicit neural representation. In fact, there exist many methods that do not incorporate the Closest Point Method face challenges related to mesh quality, and our goal is to improve these approaches."_
>
> There are no examples that support the claim that this paper improves previous approaches. A simple solver (e.g. Covector Fluids) implemented on a GPU would yield much faster results. To properly compare against previous approaches, the authors should increase the resolution of a baseline solver until it reaches a performance that is comparable to the proposed approach. I bet that a common solver would require so many variables to be that slow (on a GPU) that the application would crash with out of memory issues before reaching that point.
>
> _"The paper referred here is mainly designed to deal with the numerical dissaption caused by the adveciton and projection. But none of them are designed for the surface flow (even not with a surface flow demo). While their methods could theoretically be extended to surface flows, the additional operators they propose would require careful adaptation and design for surface-specific contexts, which is beyond the scope of this work."_
>
> Covector fluids would work on a surface mesh, they have 2D examples. That's the beauty of DEC: it really abstracts the domain representation, one has only to implement the exterior derivative properly (which is easy). I suppose that Covector Fluids did not show examples on a surface because of the limited application of such flows.
>
> _"Our approach requires the use of an additional tool based on the Closest Point Method (CPM), as described in [7], to theoretically identify the stream function and other necessary differential forms. "_
>
> CPM is not necessary to identify the streamfunction as a differential form. CPM is a way of discretizing variables, it has nothing to do with the divergence-free property or with the identification of differential forms.
>
> _"In contrast to Chorin's operator splitting scheme, our approach guarantees divergence-free behavior in a functional manner. In comparison, the advection-projection method introduces numerical diffusion, as noted in prior works (Chang et al., 2002 [8]; Elcott et al., 2007 [5]). While we employ a first-order optimizer for the non-linear problem, which may result in numerical inaccuracies, our method is still a better alternative for the classic ones both theoretically (as a form of Lie advection for divergence free field) and empirically (in the energy presevation studies)."_
>
> You are referencing papers that are **20 years old** to say **that operator splitting introduces dissipation and therefore it wont work**? What about **all the other impulse-based formulations** that were recently published? Are they all dissipative and inefficient as well? Take some time to reevaluate your perspective here.

---

> > ### Comment · Reviewer_V8TK · 2024-11-22
> >
> > _“Storage Problem. Regarding the storage concern, we acknowledge that one-step computation is often sufficient for fast local result previews. However, for studies involving long-term effects or rendering on arbitrary paths (such as in scientific research or visual effects), storage becomes a significant consideration. Additionally, for scenarios involving data sharing or long-term analysis, storage efficiency cannot be overlooked. Our model addresses these challenges by supporting continuous input and enabling to produce very high-resolution results. For comparison in the content, we believe on the current resolution is adequate to show the effectiveness and robustness of our results while higher resolution simulation for the classic method will suffer from storage and robustness problem.”_
> >
> > Storage can be a smaller problem for visual effects in the case of volumetric simulations, but I don't see the point of a method that reduces the storage for 2D simulations embedded on a 3D manifold. The performance overhead that comes with the proposed method, along  with its own computational complexity simply does not justify its usage. Moreover, I have never seen someone using surface-only fluid simulations to do even a single shot in a movie. So in my point of view, its a stretch to imply that this is going to be useful for visual effects, and the practical usability of such an approach in this context.

---

> ### Author Response · Authors · 2024-11-22
>
> We want to clarify that our goal is not to propose a general neural-based solver with streamfunctions for all flow construction and energy preservation in 2D/3D simulations. Our focus is specifically on embedded surface flow. While **time consumption** is noted as a potential limitation (similar to the time in Chen et al.), it can be mitigated through a hybrid design or more efficient representation within our framework.
>
> **Presentation** has been revised according to the suggestions from Reviewers Y5Ci and dCvp. **Evaluation** might be influenced by the non-fully convergence of the network (but still better than other baseline methods). In terms of **contribution**, while the curl of the streamfunction is not news, estimating it on a surface using a continuous functional (neural representation) could be of interest.
>
> For the **related work** section, we have included works for simulations on surfaces, which align with our previous statements, but not all the fluid simulation frameworks. Regarding boundary conditions for stream functions, our simulations are primarily on closed surfaces, and this issue does not arise. More complex surfaces (open surfaces) are discussed in Appendix F.2 as a limitation and a future research direction.
>
>
>
> The specfic problems:
> > Q1: There are no examples that support the claim that this paper improves previous approaches.
>
> > **R1:** At least, for the functional fluid on the surface (which is a classic solver), we can improve on its robustness on mesh quality as showed in Appendix. E.4. We admit the classic solver benefit much from the speed, but more efforts need to be taken on its surface implementation (including the estimation of the surface differential form) and no guarantee for the robustness of the methods on the surfaces.
>
> > Q2: Covector fluids would work on a surface mesh, they have 2D examples. That's the beauty of DEC: it really abstracts the domain representation, one has only to implement the exterior derivative properly (which is easy). I suppose that Covector Fluids did not show examples on a surface because of the limited application of such flows.
>
> > **R2:** It is not exactly the same case for 2D and 2D surface embeded in 3D. The implementation for covector fluid employ the staggered grid but not mesh for the operator estimation. Actually, accurately estimating the differential form is not a direct consequence and can introduce possible instability on the low-quality mesh (like statements in Li et al.'s  CPM poster).
>
> > Q3: CPM is not necessary to identify the streamfunction as a differential form.
>
> > **R3:** The differential form for the streamfunction does actually not need CPM but its estimation, especially continously, needs CPM (Li et al's) as a support.
>
> > Q4: Problems for impulse-based formulation.
>
> > **R4:** In our reply, we primarily address your question regarding **traditional operator splitting** on surfaces. We acknowledge that impulse-based method offers significant improvements in energy preservation and efficiency. But our main focus is the simulation on the surface with different representations. The discussion about impluse-based method in the scope needs further work on implementing differential operators and impulse gauge variables on surfaces. Actually, we hope our method to serve as a start point for integrating differential form, CPM and neural functionals (as stated in Reviewer Y5ci) via stream-function. Exploring impulse-based methods with neural representations presents an exciting future direction.
>
> > Q5: Storage Problem
>
> > **R5:** We benefit from the compact representation for the storage consumption (as noted by Reviewer Y5ci and Chen et al.). Storage can be a challenge because, even with meshes, high-resolution texture maps are needed to store simulation variables, which can be costly. In our paper, we actually aim to construct a continuous representation of surface flow, where compactness naturally reduces consumption. Several CG papers, such as those by Anzenot et al. and Elcott et al., focus on surface flow, and we follow their lead for surface visual effects. Our methods can be further optimized for time efficiency and applied into the practical scenes.

---

### Official Review · Reviewer_dCvp · 2024-11-02

**Soundness:** 4
**Presentation:** 4
**Contribution:** 4
**Rating:** 10
**Confidence:** 3

**Summary:**

The paper proposes a framework for fluid simulation on surfaces that is divergence-free by construction. This is done by using exterior calculus tools, in special the definition of the divergence based on the Hodge star operator and the exterior derivative, the property that the Hodge star is (up to a sign) its own inverse and the nilpotent property of the exterior derivative. The Closest Point Method is used to apply those tools on generalized surfaces, making a natural link with Riemannian geometry, and enabling the evaluation in the tangent space around samples in the surface. With those tools it is possible to transit between the surface and $R^3$ as needed, in special for advection which can be done considering the Riemannian metric of the manifold.

**Strengths:**

To be honest, I had a lot of fun reviewing this paper. Unless other reviewers flag major flaws that I could not find, I think it is ready for acceptance.

* It is a very good example of when a simple and elegant core idea based on strong guarantees enables a lot of very interesting questions and consequences. The core idea of using the nilpotent property of the external derivative and the self inverse property of the Hodge star operator to force a divergence free vector field on a surface is elegant and foment all the paper discussion.

* The loss formulation is as expected, very intuitive.

* As far as I know, It is the first method that converges in implicit surfaces.

* The method does not rely on training data.

* Evaluation is robust. The idea of starting with analytic examples where ground truth is easier to evaluate is good.

* Related work section cites every paper I could think of. The care with the citation of classic papers (even for datasets) is notable.

* Mathematical notation is very clean. It easy to see that there is a lot of effort with notation polishing.

* The paper makes use of very good references for background Math.

**Weaknesses:**

I will point some minor weaknesses that could be fixed to improve the paper.

(1) An image depicting equation (4) and another showing the advection process would greatly improve the friendliness of the paper since both processes are very geometric. That would make the paper be appreciated by a broader audience.

* In the image for equation (4) it is sufficient to show the neighborhood of the surface, the mapping $j$, the mapping $cp*$, the vector resulting from the gradient and the tangential vector acquired from cross product of the gradient with the normal.

* For the advection image it is sufficient to depict the push forward (pull back) function in action and the inner product using the Riemannian metric.

(2) The presentation could be more friendly by giving some intuition along the text. I will point some places I think this kind of intuition would be beneficial.

* Line 199: could say that the even though the divergence may be expressed using different k-forms, the definition of div(v) is the 0-form version resulting in a scalar function.

* Line 299 (equation 4): could say that $cp^*\sigma$ is a notation abuse because $cp^*$ expect a k-form but $\sigma$ is a 0-form. Also that the composition with $j(x)$ is to restrict the computation to the surface, the gradient is to acquire a vector field and the cross product is to acquire a tangent vector field. A reference to the proposed image would also be good here.

* Line 234 (equation 5): could say that that vorticity expression considers the rotation axis equals to the normal because it is evaluated on the surface. Then the vorticity may be represented as a scalar field.

* Line 257: could say that the expression is a neighborhood extension of the surface along the normal field.

* Line 320 (equation 13): could say that the $<. , .>_p$ notation is an inner product considering the Riemannian metric of the manifold of the tangent space at point p. A reference to the proposed image would be good here.

* Line 327 (equation 15): could say that the inner products are the first-order approximation of the push forward function.

* Line 344: that paragraph could say that the harmonic components do not contribute to the vorticity and that is the reason why the additional harmonic network is needed. Could also say that it is constant along the simulation because it is associated with the topological structure of the surface, which does not change over time.

(3) This paper deserves an acronym so it may be more easily referenced in the future by other researchers. I advise the authors to think about changing the title to include a creative acronym.

**Questions:**

(1) Why introducing $f$ in equation (8) instead of using $\sigma$ directly?

(2) I think a $t$ subscript is missing in equation (8) ($\Phi_t$).

---

> ### Author Response · Authors · 2024-11-17
>
> We deeply appreciate the reviewer's enthusiastic and high evaluation of our work, as reflected in the rare 'Strong Accept' rating. Your thoughtful and detailed feedback has greatly encouraged us, and we are genuinely grateful for your recognition. Your comments on the paper's writing are especially valuable and will greatly assist us in presenting our work more effectively.
>
> > **Q1:** An image depicting equation (4) and another showing the advection process would greatly improve the friendliness of the paper since both processes are very geometric.
>
> **R1:** We have include both figures in the revised version of the paper in Sec. 3.2 and 4.1.
>
> > **Q2:**
> > The presentation could be more friendly by giving some intuition along the text. I will point some places I think this kind of intuition would be beneficial.
>
> **R2:** We will incorporate these refinements in the revised manuscript as follows:
>
> 1. Line 199: Add the description form of the divergence function.
> 2. Line 299: Include the reference image.
> 3. Line 234: Add a description of the vorticity.
> 4. Line 320: Include a reference image for the covariant derivative.
> 5. Line 327: Relate the inner product to the first order approximation.
> 6. Line 344: Provide an explanation for harmonic component modeling and refer to Appendix F.1 for further discussion.
>
> We sincerely appreciate your efforts to help us improve the clarity and presentation of our work.
>
>
> > **Q3:** This paper deserves an acronym so it may be more easily referenced in the future by other researchers. I advise the authors to think about changing the title to include a creative acronym.
>
> **R3:** We believe NFFS (Neural Functional Flow on Surface) is sufficient and we will include it in our revised manuscript.
>
> > **Q4:** Why introducing $f$ in equation (8) instead of using $\omega$ directly?
>
> **R4:** Yes, I think we can directly use $\omega$.

---

> ### Comment · Reviewer_dCvp · 2024-11-21
>
> Thank you for the reply. I appreciate the efforts to include the presentation changes. Following reviewer Y5ci, I also respectfully disagree with reviewer V8TK for the same reasons. This paper is not only a sound technical advance, but also a very good example of proper math writing in a machine learning paper, which is somewhat difficult to find among the numerous submissions. The insights about links between exterior calculus and riemannian geometry may also be very valuable for graduate students and researchers in numerous contexts. I will champion this paper unless a major flaw is identified afterwards.

---

> > ### Author Response · Authors · 2024-11-22
> >
> > Thank you for your kind support and appreciation. We’re glad you found the technical and mathematical aspects valuable. We greatly appreciately your willingness to champion our work.

---

### Official Review · Reviewer_Gjy9 · 2024-11-03

**Soundness:** 3
**Presentation:** 2
**Contribution:** 3
**Rating:** 6
**Confidence:** 3

**Summary:**

This paper builds on the recently introduced Closest Point Exterior Calculus (CP-EC) to propose a novel method for preserving the divergence-free property of vector fields on surfaces. By leveraging the closest point map, this approach seamlessly extends computations from the surface to the surrounding Euclidean space. At the core of the paper, Theorem 3.1 presents a specific construction for generating a divergence-free vector field on a surface using the CP-EC framework. This framework enables the calculation of gradient, divergence, and curl in a way that respects the intrinsic geometry of the surface, ensuring that the velocity field remains divergence-free when constrained to the surface. A key advantage of this method is its flexibility, as it supports simulations on various surface representations, including analytic surfaces, explicitly defined mesh surfaces, and, notably, neural implicit surfaces. The paper introduces a complementary advection process based on covariant derivatives for fluid dynamics, designed to minimize energy dissipation. Numerical studies confirm the framework’s accuracy, energy preservation, memory efficiency, and adaptability to geometry. Results show it achieves about 15 times higher accuracy than other methods with similar storage, offers 5 times memory savings over classic methods, and effectively models fluid dynamics. Additionally, the simulator's robustness is demonstrated through an end-to-end generation task and a real-world velocity field decomposition.

**Strengths:**

The paper shows that the recently introduced Closest Point Exterior Calculus (CP-EC) is very well suited to simulate fluid simulation on neural implicitly defined surfaces in 3D. The CP-EC allows to automatically guarantee the divergence-free properties of the vector field. The method achieves up to 15 times higher accuracy than previously used discretization methods on the surface with the same memory requirements, which is confirmed by extensive numerical simulations of different applications.

**Weaknesses:**

The English in the current version of the paper needs to be improved. Numerous articles are missing and sometimes the wrong words are used (subtle instead of subleties, divergence free instead of divergence free property, etc).

Compared to the actual straigth forward application of the CP-EC to the case of flow simulation on surfaces, the paper seems cumbersomely long and is also not as clear to read as the recent papers on the topic referenced in the paper, whose presentation is clearer and more concise.  Maybe the authors can try to improve on that.

**Questions:**

How do you do the interpolation of the pulled forms? In the CP-EC poster the authors recommended the Cubic Lagrangian.
How does this interpolation affect the divergence-free property of the velocity field? Were there any numerical problems?
Is it possible to do an ablation study on this point?

---

> ### Author Response · Authors · 2024-11-17
>
> We thank the reviewer for the careful review and constructive advice. To address your questions and concerns:
>
> > **Q1:** The English in the current version of the paper needs to be improved.
>
> **R1:**  For any writing issues are present, we assure you that we will conduct a thorough proof reading and optimize the structure and descriptions throughout the paper.
>
>
> > **Q2:** Compared to the actual straigth forward application of the CP-EC to the case of flow simulation on surfaces, the paper seems cumbersomely long and is also not as clear to read as the recent papers on the topic referenced in the paper, whose presentation is clearer and more concise. Maybe the authors can try to improve on that.
>
> **R2:**  Thank you for your valuable advice. We include more detailed preliminaries to assist readers who may not be familiar with differential geometry. For CP-EC and divergence-free design part, the main content introduces only the necessary concepts and notion, with the proofs provided in Appendix B. We will stress the point, allowing readers with a foundation in the topic to skip the preliminaries if desired.
>
> Regarding the advection part, we believe it is essential to explain how Lie advection works, as this understanding is crucial for interpreting our loss design. We will refine the presentation to enhance clarity and ensure it aligns with the quality and precision of the referenced paper.
>
>
>
>
> > **Q3:** How do you do the interpolation of the pulled forms? In the CP-EC poster the authors recommended the Cubic Lagrangian. How does this interpolation affect the divergence-free property of the velocity field? Were there any numerical problems? Is it possible to do an ablation study on this point?
>
> **R3:** Actually, interpolation for the pull forms is not required in our approach. In the classical CP-EC method, interpolation is necessary because the functional is represented on grid points. However, in our method, we utilize a continuous network representation, where interpolation can be considered included. The numerical issues encountered in our approach might be caused by inadequate fitting rather than interpolation errors.
>
> Additionally, to elaborate further, if the classical method were used with interplation for CP-EC, divergence-free behavior would only be approximately preserved, with errors arising from the discretized operator computation.

---

### Official Review · Reviewer_Y5ci · 2024-11-04

**Soundness:** 4
**Presentation:** 4
**Contribution:** 3
**Rating:** 8
**Confidence:** 5

**Summary:**

This paper introduces a novel framework for simulating incompressible Eulerian fluid flow on 3D surfaces using neural implicit representations. This method leverages the Closest Point Method (CPM) and exterior calculus to parameterize the fluid’s velocity and vorticity fields directly on the surface without relying on discretization, which reduces memory costs and bypasses the need for conventional spatial discretization. The framework introduces a covariant-derivative-based advection process, which integrates surface flow dynamics while minimizing energy dissipation. Notably, this work is among the first to simulate incompressible fluid dynamics on neural surfaces, achieving enhanced accuracy and energy preservation across various geometric representations.

**Strengths:**

### CPM Formulation
The math formulation is clean and concise. It is quite apparent that the authors are coming from a graphics background and I love this clean DDG writing style.
- The Closest Point Method (CPM) is relatively new in visual computing, yet its integration with neural fields here aligns with my belief in CPM’s potential for solving PDEs on surfaces. Compared to surface sampling techniques (as seen in Geometry Processing with Neural Fields [Yang et al., 2021] and similar studies), CPM offers a structured way to define differential operators in volumetric data by rigorously establishing value transfer in the ambient space embedding the surface.
- A persistent challenge in neural implicit representations is that, while data is represented volumetrically (e.g., through neural SDFs), the actual solutions are constrained to the 0-level isosurface. Sampling on this isosurface can be inefficient, but CPM provides an effective alternative by leveraging the ambient space, enhancing both efficiency and rigor.

Overall, I would love to see this line of work being continued and the math formulation should be shared and seen within the ML community.

---
Some misc comments:
- The related works section is thoughtfully composed, with necessary references cited and no excess, reflecting high-quality citation practices.
- The choice of ground truth in this paper is well-justified and suitable for the presented comparisons.

**Weaknesses:**

I have two concerns, regarding the claimed first and third contributions.

---

### Performance vs. Storage vs. Accuracy
The storage and accuracy benefits presented as a core contribution appear somewhat overstated since these gains stem from the inherent compact representation of neural fields, as noted in prior works like INSR-PDE (Chen et al., 2023). The neural network, here largely a standard MLP, serves as a model reduction tool or compressed parameter space. However, the substantial cost is slower simulation speeds, particularly noticeable in evolving the PDE on a neural representation, and this tradeoff is well-documented in the field, tracing back to foundational work like *Geometry Processing with Neural Fields* (Yang et al., 2021). Additionally, working with surface PDEs inherently mitigates spatial complexity compared to volumetric Eulerian approaches, further diluting the impact of memory savings in this context. Unless optimized network designs or implementation techniques were used, this contribution may feel more like a tradeoff typical of neural fields than a novel improvement.

**TL;DR:** Without unique implementation optimizations, this tradeoff doesn’t stand out as an independent contribution, as neural networks naturally offer compact representations at the expense of computational speed.

---

### First to Simulate on Neural Implicit Surface Representation
The claim of being the first to simulate incompressible fluid flow on neural implicit surfaces is somewhat uncertain, as prior work using sampling techniques, like *Geometry Processing with Neural Fields* (Yang et al., 2021) or INSR-PDE, could also solve surface PDE like Laplace Equation by sampling on the surface. While it’s conceivable that these methods struggle with incompressibility when applied to Navier-Stokes, demonstrating their limitations would highlight the advantages of the Closest Point Method (CPM) for ensuring divergence-free constraints on neural surfaces. Including such comparative results, even as failure cases, could effectively underscore this paper’s unique approach.

**Questions:**

### Questions
1. **Use of DEC Language**: The paper’s use of Discrete Exterior Calculus (DEC) is rigorous and suits the formal approach taken. However, many in the ML and physics communities might be more accustomed to traditional differential or vector calculus, so DEC may require more adjustment for those readers. Adding intuitive explanations alongside the DEC formalism could enhance accessibility, although this may vary depending on the preferences of other reviewers.
2. **Handling Narrow Geometric Features in CPM**: The reliance on ambient space in CPM may lead to ambiguities when processing narrow or thin features. Clarifying whether this dependency impacts stability or accuracy for such geometries would enhance the framework’s applicability and inform potential adaptations to handle such cases.
---
### Suggestions
1. **Missing Citations**
    1. For by construction divergence-free field with neural network, maybe also cite [Deep Fluids](https://onlinelibrary.wiley.com/doi/10.1111/cgf.13619).
2. **Clarifying Performance Gains Over INSR**:Intuitive explanation of why your method is > INSR > PINN when constrained by storage size. Intuitively, INSR is superior to PINN because it doesn’t record time in the neural field, so, given the same storage budget, INSR should and must outperform PINN. However, your method doesn’t gain from saving less information in the neural field to achieve higher accuracy (i.e., it doesn’t concentrate model expressiveness on specific features to achieve this). So, what is the intuitive reason behind your method’s improved results over INSR? Is it due to the CPM formulation or the Helmholtz decomposition? An “ablation” would be helpful here.

---

> ### Author Response · Authors · 2024-11-17
>
> We sincerely thank the reviewer for the enthusiastic and exceptionally positive evaluation of our work. Your insightful and detailed feedback has been deeply encouraging, and we are truly grateful for your recognition and support. Your comments are highly constructive and will help us further clarify the contributions of our work.
>
> > **Q1:** Clarfications on **Performance vs. Storage vs. Accuracy**.
>
> **R1:**
> It is correct. The trade-off is effectively described in INSR-PDE, highlighting the advantages of compact neural representations. We will clarify this point in our contributions by explicitly mentioning the power of neural networks (as also referenced in Chen et al. [1]). We believe it is important to report this aspect and improvement though it is attributed to compact neural represnetation, as surface scenarios have not yet been simulated using the implicit neural representations (INR).
>
>
> > **Q2:** Clarfications on **First to Simulate on Neural Implicit Surface**
>
> **R2:**
> That is correct. First, we will restate our contribution as: ``the first study to present simulation results of incompressible fluid flow on neural implicit surfaces with a guarantee of divergence-free behavior.``
>
> Additionally, we have included more results in Appendix E.3 to further illustrate the effectiveness of our divergence-free functional. These results also demonstrate its utility in improving the surface sampling method for handling incompressibility on the analytic surface. However, for non-analytic surfaces, designing a divergence-free functional without CPM is non-trivial. Consequently, the surface sampling method would fail without a proper divergence-free design by CPM similar as the failure in analytic surfaces as shown in the Appendix E.3.
>
>
>
> > **Q3:** **Use of DEC Language**
>
> **R3:** Thanks for your advice. We follow the suggestion of Reviewer dCvp and add more illustrations to make it more intuitive for broader readers.
>
>
> > **Q4:** **Handling Narrow Geometric Features in CPM**
>
> **R4:** Yes, narrow or thin features can introduce ambiguity. It will result in
> inefficiency, inaccuracy, and energy dissipation due to increased difficulties in convergence since the network will be misleaded by the non-unique closet point mapping. To mitigate this issue, as mentioned in Appendix F.1, we can follow the approaches of (King et al. 2023 [2], Marz and Macdonald 2012 [3]) and sample the neighborhood within a smaller tube radius (closer to the surface). In parctice,
> we can determine the sampling distance threshold to the surface as less than $\Delta x$ in the equation in Section 8, guided by the estimation of the reach distance (Aamari et al. 2019 [4]). This estimation can be locally constructed using a simple mesh extractor from the signed distance function (SDF), if no mesh is available, to detect thin regions roughly and adaptively adjust the sampling distance. This adaptive approach will improve efficiency and convergence and we will include more details about adaptations in Appendix F.1.
>
>
> > **Q5:** **Missing Citations**
>
> **R5:** Thank you for pointing this out. We will include it in the related work section under the part ``Physical Simulation based on Neural Network``.
>
> > **Q6:** **Clarifying Performance Gains Over INSR**
>
> **R6:** Actually, we believe the improvement is primarily attributed to the Helmholtz decomposition.
> In the comparison between INSR and our method on the sphere jet, INSR is implemented using spherical coordinates with a fixed radius. We adopt two parameters ($\theta$, $\phi$) as input to the network, enabling the analytical verification of surface divergence in spherical coordinates without relying on CPM. Alternatively, this could also be imagined as maintaining a perfect CPM in 3D, where  the value at each 3D point is taken as the projection onto the sphere. However, even under the idealized conditions, the results still exhibit significant error.
> If an inperfect CPM is included, additional errors in operator estimation would further worsen the results. Furthermore, in Appendix E.3, we adapt our divergence-free parameterization design to a surface-sampling Eigen-Net (using Spherical Harmonics to estimate the Laplacian-Beltrami operator analytically), and the results show improvement.
>
> Nevertheless, we believe that CPM remains a critical component for handling arbitrary surfaces (especially non-analytical surfaces). It not only serves as a theoretical tool for constructing continuous differential forms to ensure divergence-free properties on arbitrary surfaces but also as a practical approach to enhances sampling efficiency by enabling uniform sampling in the ambient space, rather than directly on the iso-surface.

---

> > ### Author Response · Authors · 2024-11-17
> >
> > **Reference**:
> >
> > [1] Chen H, Wu R, Grinspun E, et al. Implicit neural spatial representations for time-dependent pdes//International Conference on Machine Learning. PMLR, 2023: 5162-5177.
> >
> > [2]. King N, Su H, Aanjaneya M, et al. A Closest Point Method for PDEs on Manifolds with Interior Boundary Conditions for Geometry Processing[J]. ACM Transactions on Graphics, 2024.
> >
> > [3]. Marz T, Macdonald C B. Calculus on surfaces with general closest point functions[J]. SIAM Journal on Numerical Analysis, 2012, 50(6): 3303-3328.
> >
> > [4]. Aamari E, Kim J, Chazal F, et al. Estimating the reach of a manifold[J]. 2019.

---

> > ### Comment · Reviewer_Y5ci · 2024-11-21
> >
> > Thank you for your reply. I found the discussion on **Q6/R6** very insightful.
> >
> > Regarding **Q2/R2**, while I understand the rationale for downplaying the contribution to address reviewer concerns, I believe it is entirely appropriate to highlight the application of CPM to neural representations. Personally, I consider CPM to be an excellent formalism for defining differential forms in modern neural representations, which I think the ML community would benefit greatly from. That said, the modified contribution is fine as it is.
> >
> > However, I respectfully disagree with reviewer V8TK’s reasoning. While it is true that there are previous works on CPM (e.g., [Li et al., 2023] and [King et al., 2024], and [Li et al., 2023] is a *poster*), I don’t believe this invalidates the significance of bringing CPM into this context. This is a unique and valuable contribution, and the reasoning used to justify such low scores seems *overly harsh*. CPM aligns naturally with modern neural representations due to their "volumetric" and "dense" nature, unlike explicit representations of CPM, which typically require a **dense grid** with spatial data structure.

---

> > > ### Author Response · Authors · 2024-11-22
> > >
> > > Thank you for your feedback. I agree that emphasizing the application of CPM to neural representations is important, as it provides valuable insights to the ML community. We again appreciate your perspective and support.

---

### Meta-Review · Area_Chair_srav · 2024-12-19

**Metareview:**

In this paper, the authors propose a neural physical simulation framework on surfaces using implicit neural representations. Their method constructs a parameterized vector field utilizing the DEC and the CPM on surfaces, achieving divergence-free simulation across different representations. Additionally, the authors adopt a covariant derivative-based advection process for surface flow dynamics and energy preservation. In their experiments, they demonstrate the lowest simulation error compared to other methods while maintaining similar memory consumption.

However, there are several limitations to the proposed method. First, the computational time cost is very high, which limits its practical applications. The examples presented in the paper are relatively simple. Moreover, given the same computational time or with a larger memory footprint -- as the current memory usage is small -- it's not clear whether existing methods could improve their accuracy to the level of the proposed method. Other surface fluid simulation methods, such as [1-2] have showed much more complex demos, but it is unclear that the proposed can achieve a similar demo as well.

Nevertheless, as the authors have stated, this is the first study to present simulation results of incompressible fluid flow on neural implicit surfaces with a theoretical guarantee of divergence-free behavior, yielding positive results. Moreover, the authors promise to improve the presentation quality of the paper. Therefore, I recommend accepting the paper but suggest it be presented as a poster.

[1] A Vortex Particle-on-Mesh Method for Soap Film Simulation, Tao et al. 2024

[2] A Moving Eulerian-Lagrangian Particle Method for Thin Film and Foam Simulation, Deng et al. 2023

**Additional Comments On Reviewer Discussion:**

There is a significant disparity among the reviewers' opinions. The reviewers who provided positive feedback recognize the paper's contribution in combining DEC and CPM with neural representation, noting that the proposed method achieves high accuracy. Conversely, Reviewer V8TK who offered negative feedback believe that the proposed method is time -- consuming even on simple examples, leading them to question its practical usefulness. Moreover, Reviewer V8TK thinks the experiments are neither comprehensive nor properly conducted, and V8TK question the technical contribution of the paper. Despite efforts by both the positive reviewers and the authors, the concerns of Reviewer V8TK remain unresolved. Given the current status of the rebuttal and the reviews, this leads to my final decision.

---

### Decision · Program_Chairs · 2025-01-22

Accept (Poster)